# Complementary cognitive roles for D2-MSNs and D1-MSNs during interval timing

Robert A Bruce[1], Matthew Weber[1], Alexandra Bova[1], Rachael Volkman[1], Casey Jacobs[1], Kartik Sivakumar[1], Hannah Stutt[1], Youngcho Kim[1], Rodica Curtu[2,3], Nandakumar S Narayanan[1,3]*

[1]Department of Neurology, University of Iowa, Iowa City, United States; [2]Department of Mathematics, University of Iowa, Iowa City, United States; [3]The Iowa Neuroscience Institute, Iowa City, United States

*For correspondence: nandakumar-narayanan@uiowa.edu

**eLife Assessment**

This **valuable** study examines the activity and function of dorsomedial striatal neurons in the estimation of time. The authors examine striatal activity as a function of time as well as the impact of optogenetic striatal manipulation on the animal's ability to estimate a time interval, providing **solid** evidence for their claims. The study could be further strengthened with a more rigorous characterization of activity and a stronger connection between their proposed model and the experimental data. The work will be of interest to neuroscientists examining how striatum contributes to behavior.

**Abstract** The role of striatal pathways in cognitive processing is unclear. We studied dorsomedial striatal cognitive processing during interval timing, an elementary cognitive task that requires mice to estimate intervals of several seconds and involves working memory for temporal rules as well as attention to the passage of time. We harnessed optogenetic tagging to record from striatal D2-dopamine receptor-expressing medium spiny neurons (D2-MSNs) in the indirect pathway and from D1-dopamine receptor-expressing MSNs (D1-MSNs) in the direct pathway. We found that D2-MSNs and D1-MSNs exhibited distinct dynamics over temporal intervals as quantified by principal component analyses and trial-by-trial generalized linear models. MSN recordings helped construct and constrain a four-parameter drift-diffusion computational model in which MSN ensemble activity represented the accumulation of temporal evidence. This model predicted that disrupting either D2-MSNs or D1-MSNs would increase interval timing response times and alter MSN firing. In line with this prediction, we found that optogenetic inhibition or pharmacological disruption of either D2-MSNs or D1-MSNs increased interval timing response times. Pharmacologically disrupting D2-MSNs or D1-MSNs also changed MSN dynamics and degraded trial-by-trial temporal decoding. Together, our findings demonstrate that D2-MSNs and D1-MSNs had opposing dynamics yet played complementary cognitive roles, implying that striatal direct and indirect pathways work together to shape temporal control of action. These data provide novel insight into basal ganglia cognitive operations beyond movement and have implications for human striatal diseases and therapies targeting striatal pathways.

## Introduction

The basal ganglia have been heavily studied in motor control (*Albin et al., 1989*). These subcortical nuclei are composed of two pathways defined by medium spiny neurons (MSNs) in the striatum:

the indirect pathway, defined by D2-dopamine receptor-expressing MSNs (D2-MSNs), and the direct pathway, defined by D1-dopamine receptor-expressing MSNs (D1-MSNs). It has been proposed that the indirect and direct pathways play opposing roles in movement (*Alexander and Crutcher, 1990*; *Cruz et al., 2022*; *Kravitz et al., 2010*), although recent work identified how these pathways can play more complex and complementary roles (*Cui et al., 2013*; *Tecuapetla et al., 2016*). MSNs receive dense input from both motor and cognitive cortical areas (*Averbeck et al., 2014*; *Graybiel, 1997*; *Middleton and Strick, 2000*), but the respective roles of D2-MSNs and D1-MSNs in cognitive processing are largely unknown. Understanding basal ganglia cognitive processing is critical for diseases affecting the striatum such as Huntington's disease, Parkinson's disease, and schizophrenia (*Andreasen, 1999*; *Hinton et al., 2007*; *Narayanan and Albin, 2022*). Furthermore, pharmacological and brain stimulation therapies directly modulate D2-MSNs, D1-MSNs, and downstream basal ganglia structures such as the globus pallidus or subthalamic nucleus. Determining basal ganglia pathway dynamics during cognitive processes will help avoid side effects of current treatments and inspire novel therapeutic directions.

We studied striatal D2-MSNs and D1-MSNs during an elementary cognitive task, interval timing, which requires estimating an interval of several seconds. Interval timing provides an ideal platform to study cognition in the striatum because timing (1) requires cognitive resources including working memory for temporal rules and attention to the passage of time *Parker et al., 2013*; (2) is reliably impaired in human striatal diseases such as Huntington's disease, Parkinson's disease, and schizophrenia *Merchant and de Lafuente, 2014*; (3) requires nigrostriatal dopamine that modulates MSNs *Emmons et al., 2017*; *Gouvêa et al., 2015*; *Matell et al., 2003*; *Mello et al., 2015*; *Monteiro et al., 2023*; *Wang et al., 2018*; and (4) can be rigorously studied in animal models (*Balci et al., 2008*; *Buhusi and Meck, 2005*; *Kim et al., 2017*; *Parker et al., 2017*; *Parker et al., 2015*). We and others have found that striatal MSNs encode time across multiple intervals by time-dependent ramping activity or monotonic changes in firing rate across a temporal interval (*Emmons et al., 2017*; *Gouvêa et al., 2015*; *Mello et al., 2015*; *Wang et al., 2018*). However, the respective roles of D2-MSNs and D1-MSNs are unknown. Past work has shown that disrupting either D2-dopamine receptors (D2) or D1-dopamine receptors (D1) powerfully impairs interval timing by increasing estimates of elapsed time (*Drew et al., 2007*; *Meck, 2006*). Similar behavioral effects were found with systemic (*Stutt et al., 2024*) or local infusion of D2 or D1 antagonists locally within the dorsomedial striatum (*De Corte et al., 2019*). These data lead to the hypothesis that D2 MSNs and D1 MSNs have similar patterns of ramping activity across a temporal interval.

We tested this hypothesis with a combination of optogenetics, neuronal ensemble recording, computational modeling, and behavioral pharmacology. We use a well-described mouse-optimized interval timing task (*Balci et al., 2008*; *Bruce et al., 2021*; *Larson et al., 2022*; *Stutt et al., 2024*; *Tosun et al., 2016*; *Weber et al., 2023*). Strikingly, optogenetic tagging of D2-MSNs and D1-MSNs revealed distinct neuronal dynamics, with D2-MSNs tending to increase firing over an interval and D1-MSNs tending to decrease firing over the same interval, similar to opposing movement dynamics (*Cruz et al., 2022*; *Kravitz et al., 2010*; *Tecuapetla et al., 2016*). MSN dynamics helped construct and constrain a four-parameter drift-diffusion computational model of interval timing, which predicted that disrupting either D2-MSNs or D1-MSNs would increase interval timing response times. Accordingly, we found that optogenetic inhibition of either D2-MSNs or D1-MSNs increased interval timing response times. Furthermore, pharmacological blockade of either D2 or D1 receptors also increased response times and degraded trial-by-trial temporal decoding from MSN ensembles. Thus, D2-MSNs and D1-MSNs have opposing temporal dynamics yet disrupting either MSN type produced similar effects on behavior. These data demonstrate how striatal pathways play complementary roles in elementary cognitive operations and are highly relevant for understanding the pathophysiology of human diseases and therapies targeting the striatum.

## Results
### Mouse-optimized interval timing
We investigated cognitive processing in the striatum using a well-described mouse-optimized interval timing task which requires mice to respond by switching between two nosepokes after a 6-s interval (*Figure 1A*; see Methods; *Balci et al., 2008*; *Bruce et al., 2021*; *Larson et al., 2022*; *Tosun et al.,*

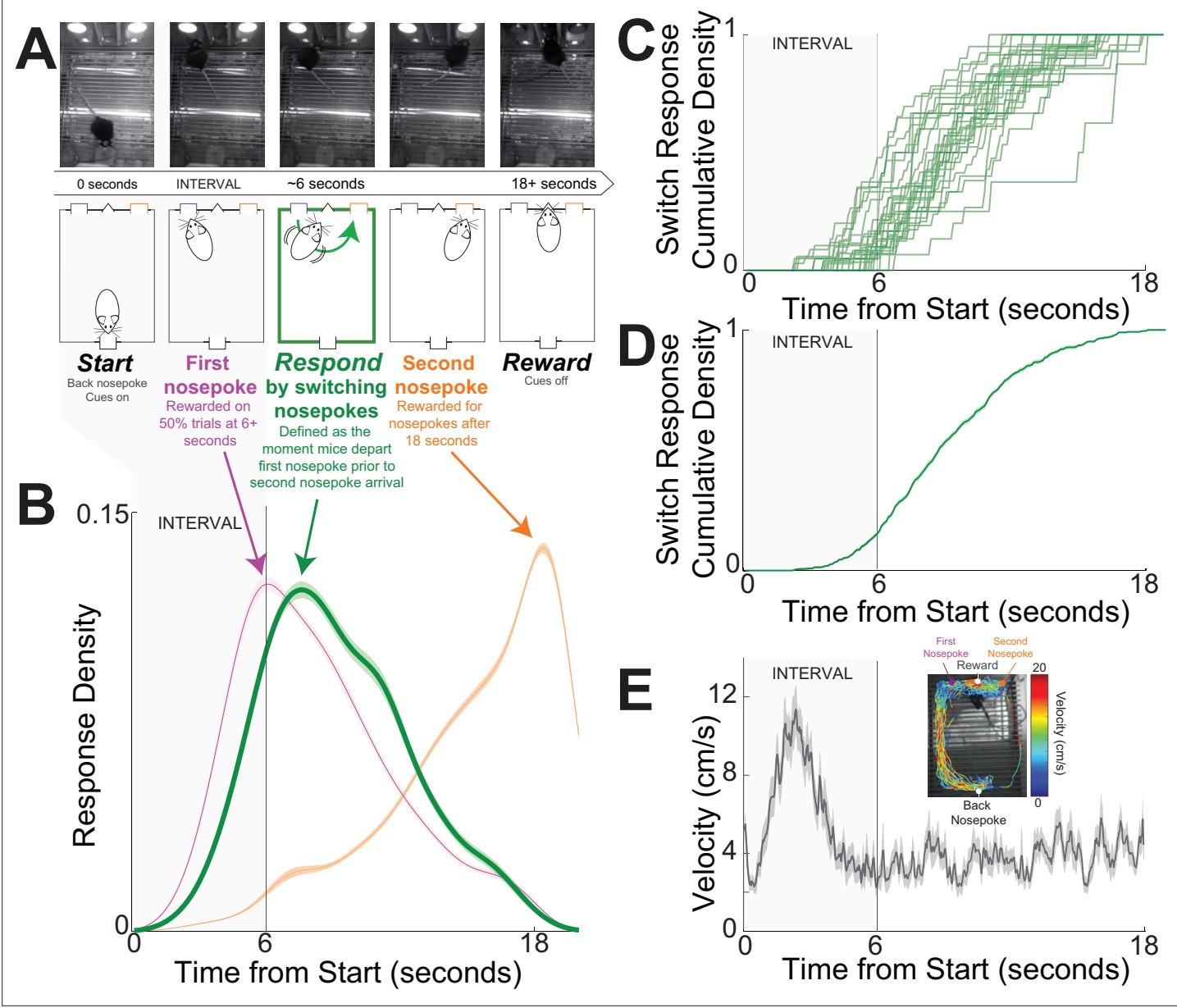

**Figure 1.** Mouse-optimized interval timing. (**A**) We trained mice to perform an interval timing task in which they had to respond by switching nosepokes after a 6-s interval (in gray shade in all panels). Mice start trials by making a back nosepoke, which triggers an auditory and visual cue. On 50% of trials, mice were rewarded for a nosepoke after 6 s at the designated 'first' front nosepoke; these trials were not analyzed. On the remaining 50% of trials, mice were rewarded for switching to the 'second' nosepoke; initial nosepokes at the second nosepoke after 18 s triggered reward when preceded by a first nosepoke. *Switch response time* was defined as the moment mice depart the first nosepoke prior to second nosepoke arrival. Because cues are identical and on for the full trial on all trials, switch responses are a time-based decision guided explicitly by temporal control of action. Indeed, mice switch nosepokes only if they do not receive a reward at the first nosepokes after the 6-s interval. Top row – screen captures from the operant chambers during a trial with switch response. (**B**) Response probability distribution from 30 mice for first nosepokes (purple), switch responses (green), and second nosepokes (orange). Responses at the first nosepoke peaked at 6 s, and switch responses peaked after 6 s. Because nosepoking at the second nosepoke was only rewarded after 18 s, second nosepokes tended to be highly skewed. Shaded area is standard error. (**C**) Cumulative switch response density for each of 30 mice. (**D**) Average cumulative switch response density; shaded area is standard error. (**E**) DeepLabCut tracking of position during interval timing from a single mouse behavioral session revealed increased velocity after trial start and then constant velocity throughout the trial. Shaded area is standard error. In (**A–E**), the 6-s interval is indicated in gray.

**Table 1.** Summary of mice, sessions, # of switch trials, and medium spiny neurons (MSNs) (medians (Q1–Q3)).

| Experiment | Figure | Cohort | Mice | # sessions | # of switch responses | Neurons |
|---|---|---|---|---|---|---|
| Interval timing behavior | *Figure 1B, C* | 1 | 30 wild-type mice | 2 (2–2) | 34 (25–42) | ~ |
| Optogenetic tagging of D2-MSNs | *Figures 2 and 3* | | 4 D2-Cre mice | 1 (1–1) | 24 (22–25) | 32 D2-MSNs |
| Optogenetic tagging of D1-MSNs | *Figures 2 and 3* | 2 | 5 D1-Cre mice | 1 (1–1) | 22 (10–28) | 41 D1-MSNs |
| Drift-diffusion models (DDM) | *Figure 4A–D* | N/A | N/A | N/A | N/A | N/A |
| Optogenetic inhibition of D2-MSNs | *Figure 5A, B* | | 10 D2-Cre mice | 6 (4–8) | 127 (78–135) | ~ |
| Optogenetic inhibition of D1-MSNs | *Figure 5C, D* | | 6 D1-Cre mice | 6 (4–8) | 80 (60–106) | ~ |
| Optogenetic D2-MSN controls | *Figure 5—figure supplement 2* | | 5 D2-Cre mice | 6 (4–7) | 103 (60–140) | ~ |
| Optogenetic D1-MSN controls | *Figure 5—figure supplement 2* | 3 | 5 D1-Cre mice | 6 (4–7) | 102 (78–128) | ~ |
| Pharmacological D2 blockade | *Figure 5E, F* | | 10 wild-type mice | 2 (2–2) | 28 (17–36) | ~ |
| Pharmacological D1 blockade | *Figure 5G, H* | 4 | | 2 (2–2) | 28 (25–30) | ~ |
| MSN ensemble recording – saline | *Figures 6 and 7* | | 4 wild-type mice, 5 D2-cre mice, and 2 D1-cre mice | 1 (1–1) | 23 (20–30) | 158 MSNs |
| MSN ensembles – D2 blockade | *Figures 6 and 7* | | | 1 (1–1) | 15 (11–22) | 167 MSNs |
| MSN ensembles – D1 blockade | *Figures 6 and 7* | 5 | | 1 (1–1) | 16 (12–26) | 144 MSNs |

*2016*; *Weber et al., 2023*). In this task, mice initiate trials by responding at a back nosepoke, which triggers auditory and visual cues for the duration of the trial. On 50% of trials, mice were rewarded for nosepoking after 6 s at the designated 'first' front nosepoke; these trials were not analyzed. On the remaining 50% of trials, mice were rewarded for nosepoking at the 'first' nosepoke and then switching to the 'second' nosepoke; initial nosepokes at the second nosepoke after 18 s triggered reward when preceded by a first nosepoke. The first nosepokes occurred before switching responses and the second nosepokes occurred much later in the interval in anticipation of reward delivery at 18 s (*Figure 1B–D*). During the task, movement velocity peaked before 6 s as mice traveled to the front nosepoke (*Figure 1E*).

We focused on the *switch response time*, defined as the moment mice exited the first nosepoke before entering the second nosepoke. Switch responses are a time-based decision guided by temporal control of action; indeed, mice switch nosepokes only if nosepoking at the first nosepoke is not rewarded after 6 s (*Figure 1B–E*). Switch responses are guided by internal estimates of time as no external cue indicates when to switch from the first to the second nosepoke (*Balci et al., 2008*; *Bruce et al., 2021*; *Tosun et al., 2016*; *Weber et al., 2023*). We defined the first 6 s after trial start as the 'interval', because during this epoch mice are estimating whether 6 s have elapsed and if they need to switch responses. In 30 mice, switch response times were 9.3 s (8.4–9.7; median (IQR)); see *Table 1* for a summary of mice, experiments, trials, and sessions. We studied dorsomedial striatal D2-MSNs and D1-MSNs using a combination of optogenetics and neuronal ensemble recordings in nine transgenic mice (four D2-Cre mice switch response time 8.2 (7.7–8.7) s; five D1-Cre mice switch response time 9.7 (7.0–10.3) s; rank sum p = 0.73; *Table 1*).

## Opposing D2-MSN and D1-MSN dynamics

Striatal neuronal populations are largely composed of MSNs expressing D2- or D1-dopamine receptors. We optogenetically tagged D2-MSNs and D1-MSNs by implanting optrodes in the dorsomedial striatum and conditionally expressing channelrhodopsin (ChR2; *Figure 2—figure supplement 1A*) in four D2-Cre (two female) and five D1-Cre transgenic mice (two female). This approach expressed ChR2 in D2-MSNs or D1-MSNs, respectively (*Figure 2A, B*; *Kim et al., 2017*). We identified D2-MSNs or D1-MSNs by their response to brief pulses of 473 nm light; neurons that fired within 5 ms were considered optically tagged putative D2-MSNs (*Figure 2—figure supplement 1B, C*). We tagged 32 putative D2-MSNs and 41 putative D1-MSNs in a single recording session during interval timing. There

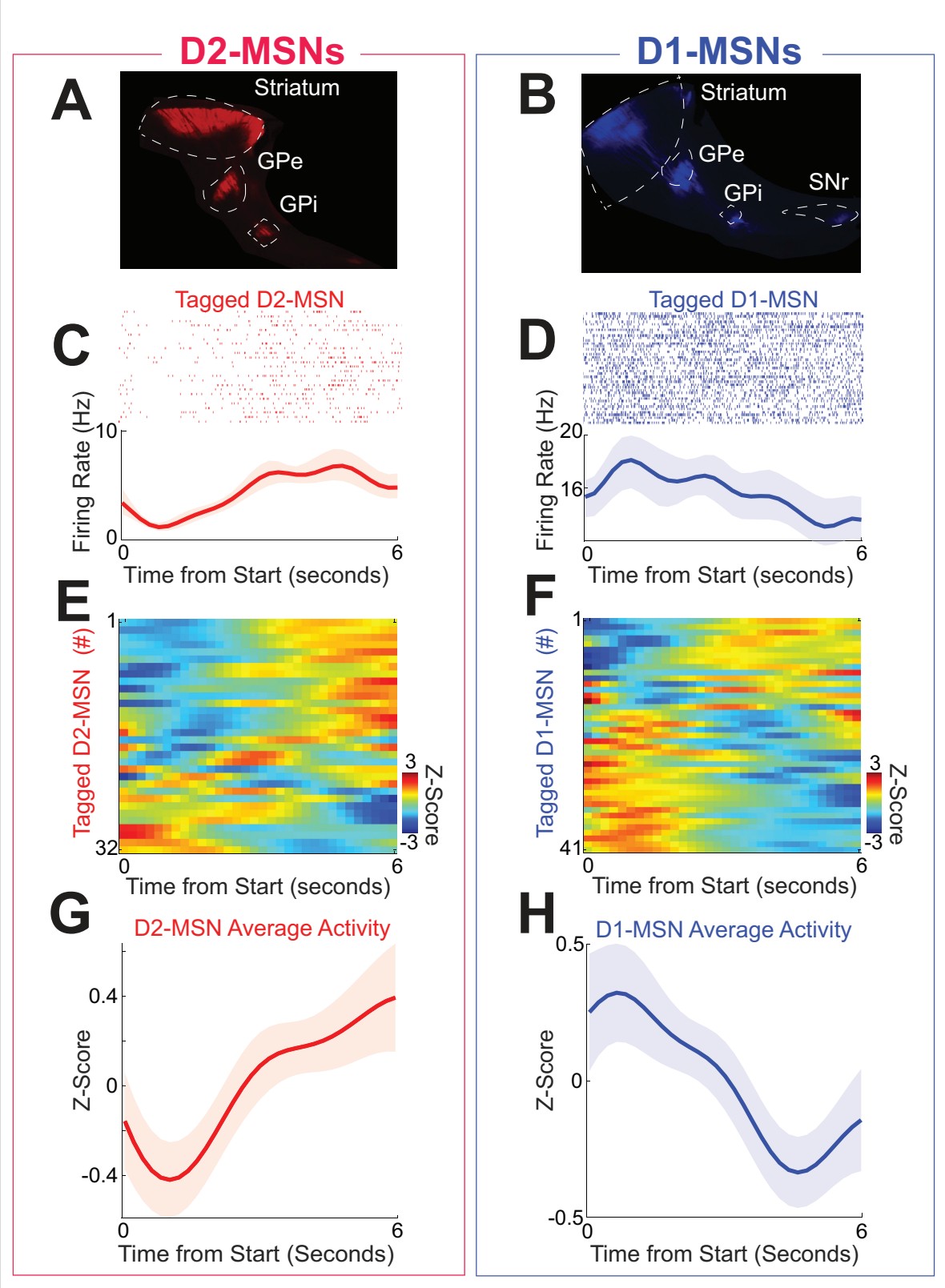

**Figure 2.** D2-MSNs and D1-MSNs have opposing dynamics during interval timing. (**A**) D2-MSNs in the indirect pathway, which project from the striatum to the globus pallidus external segment (GPe; sagittal section) and internal segment (GPi) and (**B**) D1-MSNs, which project from the striatum to the GPe, GPi, and substantia nigra (SNr; sagittal section). Peri-event raster (**C**) from an optogenetically tagged putative D2-MSN (red) and (**D**) from an optogenetically tagged putative D1-MSN (blue). Shaded area is the bootstrapped 95% confidence interval. (**E**) Peri-event time histograms (PETHs) from

*Figure 2 continued*

all D2-MSNs and (**F**) from all D1-MSNs were binned at 0.2 s, smoothed using kernel-density estimates using a bandwidth of 1, and *z*-scored. Average activity from PETHs revealed that (**G**) D2-MSNs (red) tended to ramp up, whereas (**H**) D1-MSNs (blue) tended to ramp down. Shaded area is standard error. Data from 32 tagged D2-MSNs in 4 D2-Cre mice and 41 tagged D1-MSNs in 5 D1-Cre mice.

The online version of this article includes the following figure supplement(s) for figure 2:

**Figure supplement 1.** Striatal MSN recording.

**Figure supplement 2.** D2- and D1-MSN activity over a longer epoch from 10 s prior to trial start, when mice initiated trials at the back nosepoke, to the end of 18 s, after which making a second nosepoke led to reward.

were no consistent differences in overall firing rate between D2-MSNs and D1-MSNs (D2-MSNs: 3.4 (1.4–7.2) Hz; D1-MSNs 5.2 (3.1–8.6) Hz; $F = 2.7$, p = 0.11; all neuronal analyses account for variance between mice using linear mixed-effects models). Peri-event rasters and histograms from a tagged putative D2-MSN (*Figure 2C*) and from a tagged putative D1-MSN (*Figure 2D*) demonstrate prominent modulations for the first 6 s of the interval after trial start. *Z*-scores of average peri-event time histograms (PETHs) from 0 to 6 s after trial start for each putative D2-MSN are shown in *Figure 2E* and for each putative D1-MSN in *Figure 2F*. These PETHs revealed that for the 6-s interval immediately after trial start, many putative D2-MSN neurons appeared to ramp up while many putative D1-MSNs appeared to ramp down. For 32 putative D2-MSNs average PETH activity increased over the 6-s interval immediately after trial start, whereas for 41 putative D1-MSNs, average PETH activity decreased. Accordingly, D2-MSNs and D1-MSNs had differences in activity early in the interval (0–5 s; $F = 4.5$, p = 0.04 accounting for variance between mice) but not late in the interval (5–6 s; $F = 1.9$, p = 0.17 accounting for variance between mice). Examination of a longer interval of 10 s before to 18 s after trial start revealed the greatest separation in D2-MSN and D1-MSN dynamics during the 6-s interval after trial start (*Figure 2—figure supplement 2*). Strikingly, these data suggest that D2-MSNs and D1-MSNs display distinct dynamics during interval timing.

## Differences between D2-MSNs and D1-MSNs

To quantify differences between D2-MSNs vs D1-MSNs in *Figure 2G, H*, we turned to principal component analysis (PCA), a data-driven tool to capture the diversity of neuronal activity (*Kim et al., 2017*). Work by our group and others has uniformly identified PC1 as a linear component among corticostriatal neuronal ensembles during interval timing (*Bruce et al., 2021*; *Emmons et al., 2020*; *Emmons et al., 2019*; *Emmons et al., 2017*; *Kim et al., 2017*; *Narayanan et al., 2013*; *Narayanan and Laubach, 2009*; *Parker et al., 2014*; *Wang et al., 2018*). We analyzed PCA calculated from all D2-MSN and D1-MSN PETHs over the 6-s interval immediately after trial start. PCA identified time-dependent ramping activity as PC1 (*Figure 3A*), a key temporal signal that explained 54% of variance among tagged MSNs (*Figure 3B*; variance for PC1 p = 0.009 vs 46 (44–49)% for any pattern of PC1 variance derived from random data; *Narayanan, 2016*). Consistent with population averages from *Figure 2G, H*, D2-MSNs and D1-MSNs had opposite patterns of activity with negative PC1 scores for D2-MSNs and positive PC1 scores for D1-MSNs (*Figure 3C*; PC1 for D2-MSNs: –3.4 (–4.6 to 2.5); PC1 for D1-MSNs: 2.8 (–2.8 to 4.9); $F = 8.8$, p = 0.004 accounting for variance between mice; *Figure 3—figure supplement 1*; Cohen's d = 0.7; power = 0.80; no reliable effect of sex ($F = 0.4$, p = 0.51) or switching direction ($F = 1.7$, p = 0.19)). Importantly, PC1 scores for D2-MSNs were significantly less than 0 (signrank D2-MSN PC1 scores vs 0: p = 0.02), implying that because PC1 ramps down, D2-MSNs tended to ramp up. Conversely, PC1 scores for D1-MSNs were significantly greater than 0 (signrank D1-MSN PC1 scores vs 0: p = 0.05), implying that D1-MSNs tended to ramp down. Thus, analysis of PC1 in *Figure 3A–C* suggested that D2-MSNs (*Figure 2G*) and D1-MSNs (*Figure 2H*) had opposing ramping dynamics.

To interrogate these dynamics at a trial-by-trial level, we calculated the linear slope of D2-MSN and D1-MSN activity over the first 6 s of each trial using generalized linear modeling (GLM) of effects of time in the interval vs trial-by-trial firing rate (*Latimer et al., 2015*). Note that this analysis focuses on each trial rather than population averages as in *Figures 2G, H and 3A–C*. Nosepokes were included as a regressor for movement. GLM analysis also demonstrated that D2-MSNs had significantly different slopes (–0.01 spikes/s (–0.10 to 0.10)), which were distinct from D1-MSNs (–0.20 (–0.47 to –0.06; *Figure 3D*; $F = 8.9$, p = 0.004 accounting for variance between mice; *Figure 3—figure supplement*

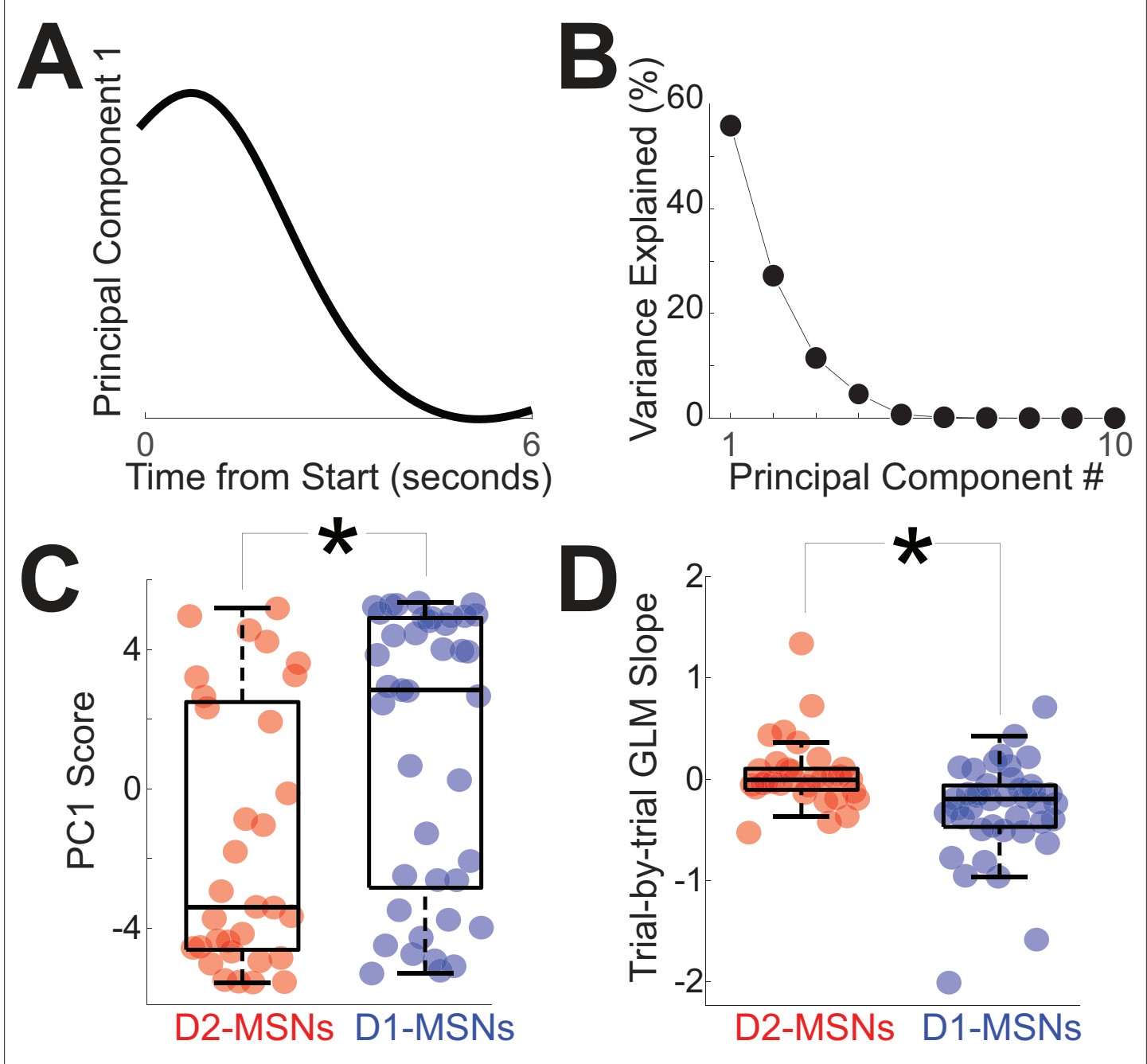

**Figure 3.** Quantification of opposing D2-MSN and D1-MSN dynamics. (**A**) Principal component analysis revealed that the first component (PC1) exhibited time-dependent ramping. (**B**) The first principal component explained ~54% of variance across tagged medium spiny neuron (MSN) ensembles. (**C**) Differences between D2-MSNs (red) and D1-MSNs (blue) were captured by PC1 which exhibited time-dependent ramping. (**D**) These differences were also apparent in the linear slope of firing rate vs time in the interval, with D1-MSNs (blue) having a more negative slope than D2-MSNs (red). In C and D, each point represents data from a tagged MSN. *p < 0.05 via linear mixed effects models accounting for variance between mice. Data from 32 tagged D2-MSNs in 4 D2-Cre mice and 41 tagged D1-MSNs in 5 D1-Cre mice.

The online version of this article includes the following figure supplement(s) for figure 3:

**Figure supplement 1.** PC1 and slopes from individual mice.

*1*); Cohen's *d* = 0.8; power = 0.98; no reliable effect of sex (*F* = 0.0, p = 0.88) or switching direction (*F* = 1.7, p = 0.19)). We found that D2-MSNs and D1-MSNs had significantly different slopes even when excluding outliers (four outliers excluded outside of 95% confidence intervals; *F* = 7.5, p = 0.008 accounting for variance between mice) and when the interval was defined as the time between trial start and the switch response on a trial-by-trial basis for each neuron (*F* = 4.3, p = 0.04 accounting for variance between mice). Trial-by-trial GLM slope was strongly correlated with PC1 scores in *Figure 3A–C* (PC1 scores vs GLM slope *r* = –0.60, p = $10^{-8}$), explaining 37% of variance. These data demonstrate that D2-MSNs and D1-MSNs had distinct slopes of firing rate across the interval and were consistent with analyses of average activity and PC1, which exhibited time-related ramping.

Our findings could not be easily explained by movement because (1) only 25% of switch responses occurred before 6 s (*Figure 1B, C*), (2) our GLM included a regressor accounting for the nosepokes when present, and (3) nosepoke GLM βs were not reliably different between D2-MSNs and D1-MSNs (*F* = 1.5, p = 0.22 accounting for variance between mice). In summary, analyses of average activity, PC1, and trial-by-trial firing-rate slopes over the interval provide convergent evidence that D2-MSNs and D1-MSNs had distinct dynamics during interval timing. These data provide insight into temporal processing by striatal MSNs.

## Drift-diffusion models of opposing D2-MSN and D1-MSN dynamics

Our analysis of average activity (*Figure 2G, H*) and PC1 (*Figure 3A–C*) suggested that D2-MSNs and D1-MSNs might have opposing dynamics. However, past computational models of interval timing have relied on drift-diffusion dynamics that increases over the interval and accumulates evidence over time (*Nguyen et al., 2020*; *Simen et al., 2011*). To reconcile how these complementary MSNs dynamics might effect temporal control of action, we constructed a four-parameter drift-diffusion model (DDM). Our goal was to construct a DDM inspired by average differences in D2-MSNs and D1-MSNs that predicted switch response time behavior. We constructed a DDM where *x* represents the neuronal firing rate of an 'output' unit collecting evidence on the activity of striatal D2-MSNs or D1-MSNs, *t* represents time measured in seconds, and *dx* and *dt* mean 'change' in *x* and *t*, respectively (equivalent to the derivative *dx/dt*):

$$dx = (F - x)\, D dt + \sigma d\xi\,(t) \tag{1}$$

$$x\,(0) = b \tag{2}$$

The model has four independent parameters $F, D, \sigma$, and $b$ (described below) and a threshold value *T* defined by

$$T = T\,(F, b) = \left(1 - \frac{1}{4}b\right) F + \frac{1}{4}b\,(1 - F) \tag{3}$$

The firing rate is set initially at baseline *b* (see *Equation 2*), then driven by input *F* akin to the dendritic current induced by the overlap of corticostriatal postsynaptic potentials (*Shepherd, 2013*). With each unit of time *dt*, we suppose there is corticostriatal stimulation that provides incremental input proportional to the activity itself, $(F - x)\, D$, to reach a decision. *D* is *the drift rate* for the event sequence and can be interpreted as a parameter inversely proportional to the neural activity's integration time constant. The drift, together with noise $\xi\,(t)$ (of zero mean and strength $\sigma$), leads to fluctuating accumulation which eventually crosses a threshold *T* (see *Equation 3*; *Figure 4A, B*). The time $t^*$ it takes the firing rate to reach the threshold, $x\,(t^*) = T$, is the *switch response time*.

Our model aimed to fit statistical properties of mouse behavioral responses while incorporating MSN network dynamics. The model does not attempt, however, to fit individual neurons' activity, because our model predicts a single behavioral parameter – switch time – that can be caused by a diversity of neuronal activity. We first analyzed trial-based aggregated activity of MSN recordings from each mouse $(x_j\,(t))$ where $j = 1, \ldots, N$ neurons. For D2-MSN or D1-MSN ensembles of $N > 11$, we found linear combinations of their neuronal activities, with some $\beta_j$ coefficients,

$$x\,(t) = \sum_{j=1}^{N} \beta_j\, x_j\,(t)$$

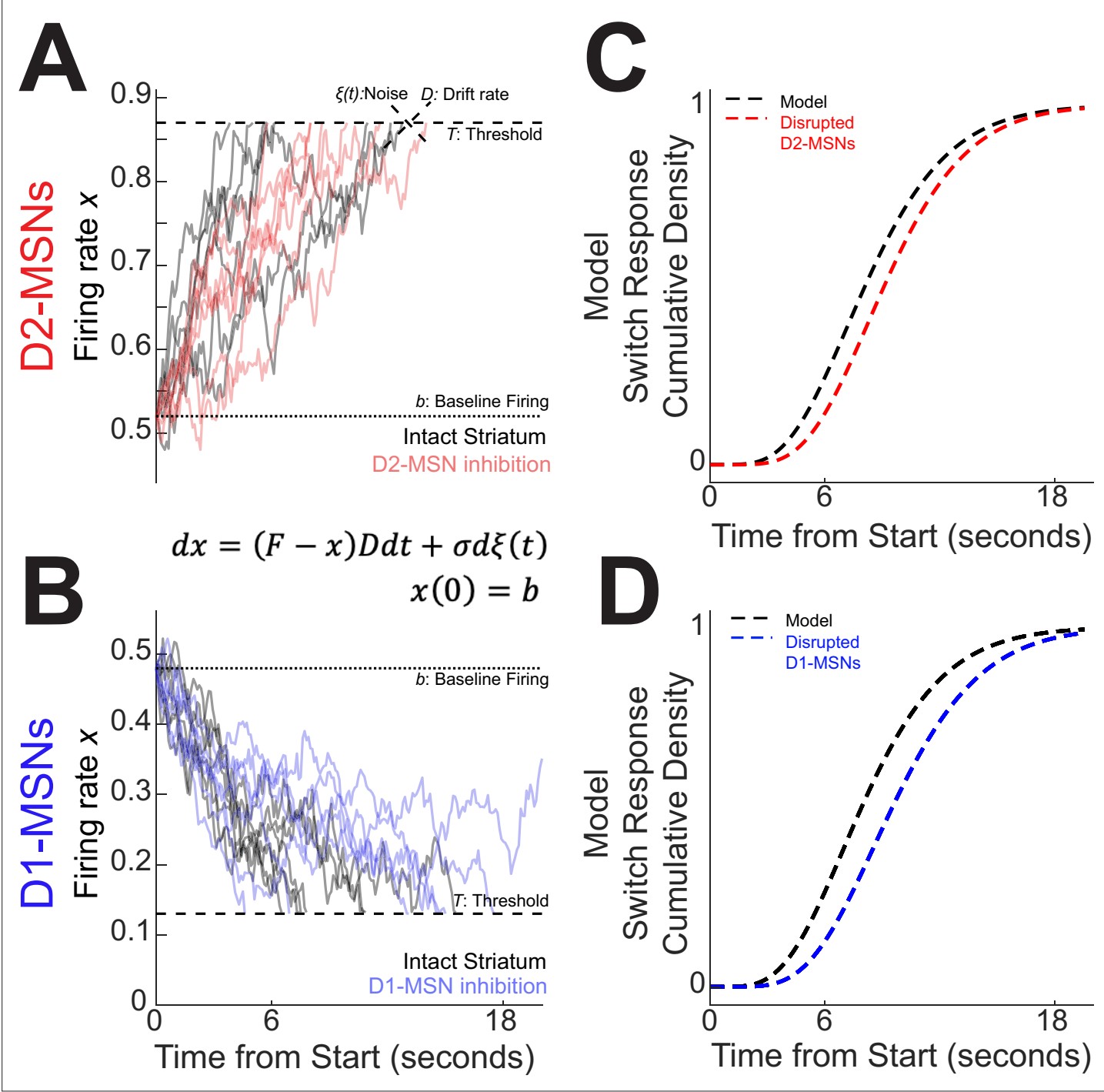

**Figure 4.** Four-parameter drift-diffusion computational model of striatal activity during interval timing. (**A**) We modeled interval timing with a low parameter diffusion process with a drift rate $D$, noise $\xi(t)$, and a baseline firing rate $b$ that drifts toward a threshold $T$ indicated by dotted lines. With D2-MSNs disrupted (solid red curves), this drift process decreases and takes longer to reach the threshold. (**B**) The same model also accounted for D1-MSNs with an opposite drift. With D1-MSNs disrupted (solid blue curves), the drift process again takes longer to reach the threshold. Because both D2-MSNs and D1-MSNs contribute to the accumulation of temporal evidence, this model predicted that (**C**) disrupting D2-MSNs would increase response times during interval timing (dotted red line) and (**D**) disrupting D1-MSNs would also increase response times (dotted blue line). Threshold $T$ depends on $b$ and target firing $F$. For details on the selection of parameter values in drift-diffusion model (DDM), see Methods and *Figures 1–3*.

The online version of this article includes the following figure supplement(s) for figure 4:

**Figure supplement 1.** Trial-by-trial predictions of switch response time from D2-MSN and D1-MSN ensemble dynamics.

*Figure 4 continued on next page*

**Figure supplement 2.** Drift-diffusion model (DDM) parameter exploration.

**Figure supplement 3.** DDM details.

that could predict the trial-by-trial switch response times (accuracy >90%, ***Figure 4—figure supplement 1***; compared with <20% accuracy for Poisson-generated spikes of same trial-average firing rate). The predicted switch time $t^*_{pred}$ was defined by the time when the weighted ensemble activity $x(t)$ first reached the value $x(t^*_{pred}) = 0.5$. Finally, we built DDMs to account for this opposing trend (increasing vs decreasing) of MSN dynamics and for ensemble threshold behavior defining $t^*_{pred}$; see the resulting model (***Equations 1–3***) and its simulations (***Figure 4A, B***).

This instantiation of DDMs captured the complimentary D2-MSN and D1-MSN dynamics that we discovered in our optogenetic tagging experiments (***Figure 2G, H*** vs ***Figure 4A, B***). The model's parameters were chosen to fit the distribution of switch response times: $F = 1, b = 0.52$ (so $T = 0.87, D = 0.135, \sigma = 0.052$) for intact D2-MSNs (***Figure 4A***, in black); and $F = 0, b = 0.48$ (so $T = 0.12, D = 0.141, \sigma = 0.052$) for intact D1-MSNs (***Figure 4B***, in black). See Methods and ***Figure 4—figure supplement 2*** for how the model parameters were chosen and how we quantified the model's explanatory power for behavior. Interestingly, we observed that two-parameter gamma distributions strongly accounted for the model dynamics (D2-MSNs: Gamma parameters $\alpha = 6.08, \beta = 0.69, R^2$ Gamma vs Model = 0.99; D1-MSNs $\alpha = 5.89, \beta = 0.69; R^2$ Gamma vs Model = 0.99; ***Figure 4C, D***, dotted black lines; ***Figure 4—figure supplement 3***). Gamma distributions also provided a surprisingly good approximation for the probability distribution and cumulative distribution functions of mouse switch response times (***Figure 1B–D***; $R^2$ Data vs Gamma = 0.94 for D1-MSNs and for D2-MSNs). Thus, in the next sections, we used gamma distributions as a proxy for both the distribution of times generated by the model and the distribution of mice switch response times, when comparing those for the goodness of fit. Our model provided the opportunity to computationally explore the consequences of disrupting D2-MSNs or D1-MSNs. Because both D2-MSNs and D1-MSNs accumulate temporal evidence, disrupting either MSN type in the model changed the slope. The results were obtained by simultaneously decreasing the drift rate $D$ (equivalent to lengthening the neurons' integration time constant) and lowering the level of network noise $\sigma$: $D = 0.129, \sigma = 0.043$ for D2-MSNs in ***Figure 4A*** (in red; changes in noise had to accompany changes in drift rate to preserve switch response time variance. See Methods); and $D = 0.122, \sigma = 0.043$ for D1-MSNs in ***Figure 4B*** (in blue). The model predicted that disrupting either D2-MSNs or D1-MSNs would degrade the temporal accumulation of evidence, increase switch response times (***Figure 4C, D***), and shift MSN dynamics. In the next section, we interrogated these ideas with a combination of optogenetics, behavioral pharmacology, and electrophysiology.

## Disrupting D2-MSNs or D1-MSNs increases switch response times

DDMs captured opposing MSN dynamics and predicted that disrupting either D2-MSNs or D1-MSNs should slow temporal processing and increase switch response times (***Figure 4***). We tested this idea with optogenetics. We bilaterally implanted fiber optics and virally expressed the inhibitory opsin halorhodopsin in the dorsomedial striatum of 10 D2-Cre mice (5 female) to inhibit D2-MSNs (***Figure 5***; fiber optic locations in ***Figure 5—figure supplement 1***; this group of mice was entirely separate from the optogenetic tagging mice). We found that D2-MSN inhibition reliably increased switch response times (***Figure 5A, B***; Laser Off: 8.6 s (8.3–9.3); Laser On: 10.2 s (9.4–10.2); signed rank p = 0.002, Cohen's $d = 1.7$). To control for heating and nonspecific effects of optogenetics, we performed control experiments in D2-cre mice without opsins using identical laser parameters; we found no reliable effects for opsin-negative controls (***Figure 5—figure supplement 2***). Remarkably, DDM predictions were highly concordant with D2-MSN inhibition behavioral data ($R^2$ Data vs Model = 0.95; ***Figure 4—figure supplement 3***) as well as with behavioral data from laser off trials ($R^2$ Data vs Model = 0.94; parameters chosen to fit laser off trials; ***Figure 4—figure supplement 3***).

Next, we investigated D1-MSNs (***Kravitz et al., 2010***). In six D1-Cre mice (three female), optogenetic inhibition of dorsomedial striatal D1-MSNs increased switch response times (***Figure 5C, D***; Laser Off: 8.7 (8.0–9.1) s; Laser On: 10.5 (9.6–11.1) s; signed rank p = 0.03, Cohen's $d = 1.4$). As with D2-MSNs, we found no reliable effects with opsin-negative controls in D1-MSNs (figure supplement 8). DDM predictions again were highly concordant with D1-MSN inhibition behavioral data $R^2$ Data

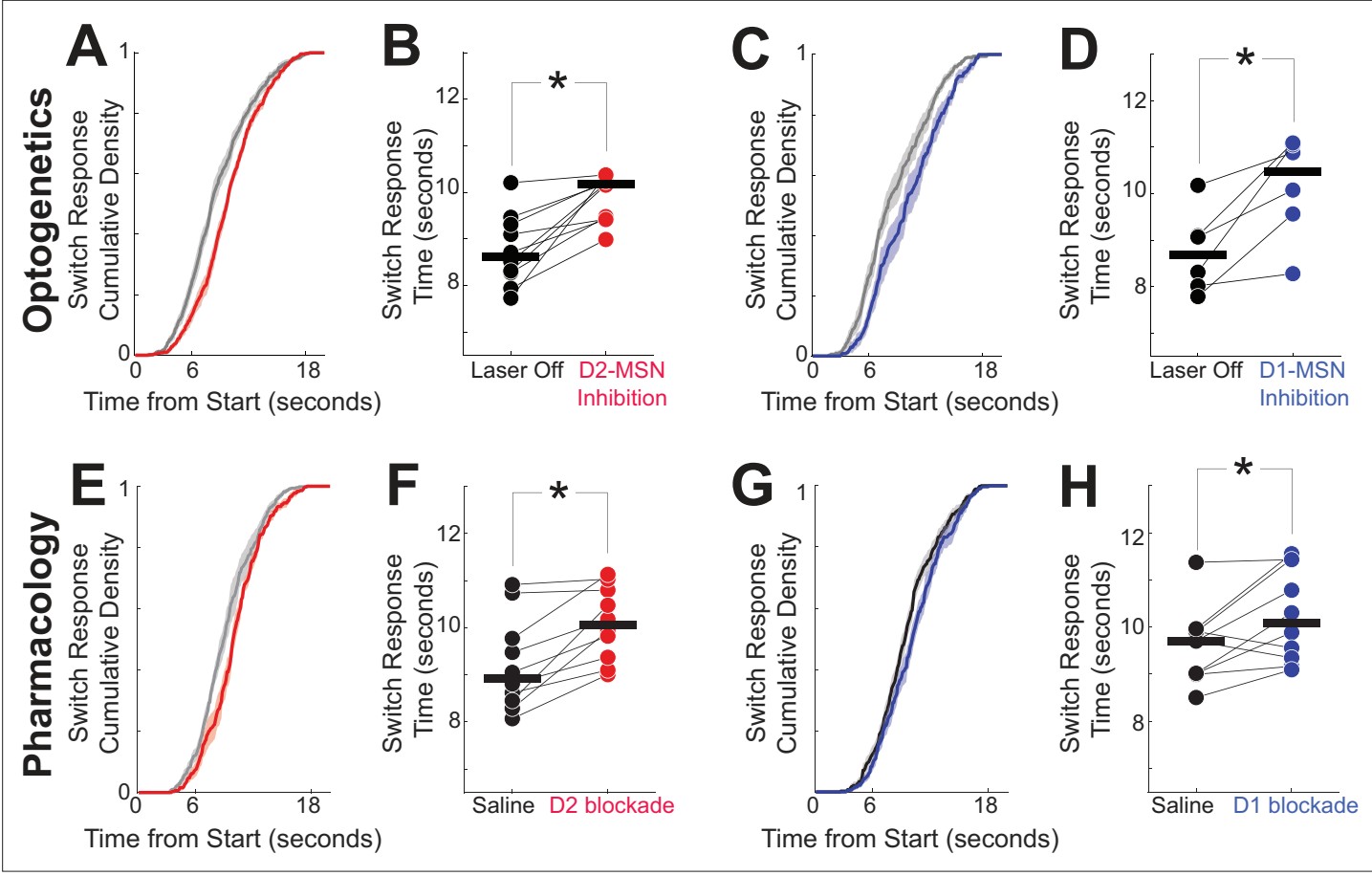

**Figure 5.** Disrupting D2- or D1-MSNs increases response times. (**A**) As predicted by our drift-diffusion model (DDM) in **Figure 4**, optogenetic inhibition of D2-MSNs (red) shifted cumulative distributions of response times to the right, and (**B**) increased response times; data from 10 D2-Cre mice expressing halorhodopsin (Halo). Also as predicted by our DDM, (**C**) optogenetic inhibition of D1-MSNs shifted cumulative distribution functions to the right, and (**D**) increased response times; data from 6 D1-Cre mice expressing Halo. Similarly, (**E**) pharmacologically disrupting D2-dopamine receptors (red) with the D2 antagonist sulpiride shifted cumulative distribution functions to the right, and (**F**) increased response times; data from 10 wild-type mice. Also, (**G**) pharmacologically disrupting D1-dopamine receptors (blue) with the D1 antagonist SCH23390 shifted cumulative distribution functions to the right, and (**H**) increased response times; data from the same 10 wild-type mice as in (**E, F**). In (**B**, **D**, **F**, and **H**) connected points represent the mean response time from each animal in each session, and horizontal black lines represent group medians. *p = <0.05, signed rank test. See **Figure 5—figure supplement 2** for data from opsin-negative controls.

The online version of this article includes the following figure supplement(s) for figure 5:

**Figure supplement 1.** Dorsomedial striatal optogenetics.

**Figure supplement 2.** Optogenetic control data.

**Figure supplement 3.** Optogenetically inhibiting D2-MSNs or D1-MSNs does not affect task-specific motor control.

**Figure supplement 4.** Switch response variance and reward rate.

vs Model = 0.95; parameters chosen to fit D1-MSN inhibition trials andwith laser off trials ($R^2$ Data vs Model = 0.94; **Figure 4—figure supplement 3**).

To test the generality of the effects observed in the optogenetic inhibition experiment, we turned to systemic pharmacology. Notably, our past work demonstrates that both systemic and local striatal D2 blockade powerfully affected interval timing (**De Corte et al., 2019**; **Stutt et al., 2024**). We used sulpiride (12.5 mg/kg intraperitoneally (IP)) to block D2-dopamine receptors in 10 wild-type mice, which increased switch response times relative to saline sessions (median (IQR); saline: 8.9 (8.4–9.8) s; D2 blockade: 10.1 (9.4–10.8) s; signed rank p = 0.002, Cohen's *d* = 1.0; **Figure 5E, F**). There was no difference in switch response time between saline sessions and laser-off trials from D2-MSN opto-genetic inhibition in D2-Cre mice, implying that in optogenetic sessions, D2-MSN inhibition did not disrupt behavior during laser-off trials (rank sum p = 0.31). These data are consistent with past work

demonstrating that disrupting dorsomedial D2-MSNs slows timing (*Drew et al., 2007*; *Stutt et al., 2024*).

In the same 10 wild-type mice, systemic drugs blocking D1-dopamine receptors (D1 blockade; SCH23390 0.05 mg/kg IP) increased switch response times (saline: 9.7 s (9.0–9.9); D1 blockade: 10.1 s (9.3–11.4); signed rank p = 0.04, Cohen d = 0.7; *Figure 5G, H*), as in our past work with both systemic and local D1 blockade (*De Corte et al., 2019*; *Stutt et al., 2024*). Once again, there was no difference between saline sessions and D1-MSN inhibition laser-off trials in D1-Cre mice (rank sum p = 0.18). These results agree with our DDM predictions and demonstrate that disrupting either D2-MSNs or D1-MSNs increases switch response time.

We found no evidence that inhibiting D2-MSNs in the dorsomedial striatum changed task-specific movements such as nosepoke duration (i.e., time of nosepoke entry to exit; *Figure 5—figure supplement 3*) or switch traversal time between the first and second nosepokes (*Figure 5—figure supplement 3*). Similarly, we found no evidence that D1-MSN inhibition changed nosepoke duration (*Figure 5—figure supplement 2*) or traversal time (*Figure 5—figure supplement 2*). Furthermore, disrupting D2-MSNs or D1-MSNs did not change switch response time standard deviations or the number of rewards (*Figure 5—figure supplement 4*). Our data suggest that disrupting D2-MSNs and D1-MSNs specifically slowed interval timing without consistently changing task-specific movements. Together, our findings suggest that optogenetically or pharmacologically disrupting dorsomedial striatal D2-MSNs and D1-MSNs degraded the accumulation of temporal evidence shown in *Figure 4*, resulted in increased switch response times.

## D2 blockade and D1 blockade shift MSN dynamics

MSN ensembles strongly encode time (*Bruce et al., 2021*; *Emmons et al., 2017*; *Gouvêa et al., 2015*; *Mello et al., 2015*; *Wang et al., 2018*), but it is unknown how disruptions in D2- or D1-dopamine receptors affect these ensembles (*Yun et al., 2023*). Although nonspecific, pharmacological experiments have two advantages over optogenetics for recording experiments: (1) many clinically approved drugs target dopamine receptors, and (2) they are more readily combined with recordings than optogenetic inhibition, which silences large populations of MSNs. We recorded from dorsomedial striatal MSN ensembles in 11 separate mice during sessions with saline, D2 blockade with sulpiride, or D1 blockade with SCH23390 (*Figure 6A*; *Figure 6—figure supplements 1–3*; data from one recording session each for saline, D2 blockade, and D1 blockade during interval timing).

We analyzed 99 MSNs in sessions with saline, D2 blockade, and D1 blockade. We matched MSNs across sessions based on waveform and interspike intervals; waveforms were highly similar across sessions (correlation coefficient between matched MSN waveforms: saline vs D2 blockade $r$ = 1.00 (0.99–1.00 rank sum vs correlations in unmatched waveforms p = 3 × 10$^{-44}$; waveforms; saline vs D1 blockade $r$ = 1.00 (1.00–1.00), rank sum vs correlations in unmatched waveforms p = 4 × 10$^{-50}$). There were no consistent changes in MSN average firing rate with D2 blockade or D1 blockade (saline: 5.2 (3.3–8.6) Hz); D2 blockade 5.1 (2.7–8.0) Hz; $F$ = 1.1, p = 0.30 accounting for variance between MSNs; D1 blockade 4.9 (2.4–7.8) Hz; $F$ = 2.2, p = 0.14 accounting for variance between MSNs; *Figure 6—figure supplement 2*).

We noticed differences in MSN activity across the interval with D2 and D1 blockades at the individual MSN level (*Figure 6B–D*) as well as at the population level (*Figure 6E*). We used PCA to quantify effects of D2 or D1 blockade (*Bruce et al., 2021*; *Emmons et al., 2017*; *Kim et al., 2017*). We constructed principal components (PCs) from $z$-scored PETHs of firing rate from saline, D2 blockade, and D1 blockade sessions for all mice together. The first component (PC1), which explained 54% of neuronal variance, exhibited 'time-dependent ramping', or monotonic changes over the 6-s interval immediately after trial start (*Figure 6F, G*; variance for PC1 p = 0.001 vs 46 (45–47)% for any pattern of PC1 variance in random data; *Narayanan, 2016*). As with the optogenetic tagging dataset, trial-by-trial GLM slopes were strongly correlated with PC1 scores (PC1 scores vs GLM slope $r$ = –0.66, p = 10$^{-14}$), explaining 44% of variance. Interestingly, PC1 scores shifted with D2 blockade (*Figure 6H*; PC1 scores for D2 blockade: –0.6 (–3.8 to 4.7) vs saline: –2.3 (–4.2 to 3.2), $F$ = 5.1, p = 0.03 accounting for variance between MSNs; no reliable effect of sex ($F$ = 0.2, p = 0.63) or switching direction ($F$ = 2.8, p = 0.10)). PC1 scores also shifted with D1 blockade (*Figure 6H*; PC1 scores for D1 blockade: –0.0 (–3.9 to 4.5), $F$ = 5.8, p = 0.02 accounting for variance between MSNs; no reliable effect of sex ($F$ = 0.0, p = 0.93) or switching direction ($F$ = 0.9, p = 0.34)). There were no reliable differences in PC1 scores

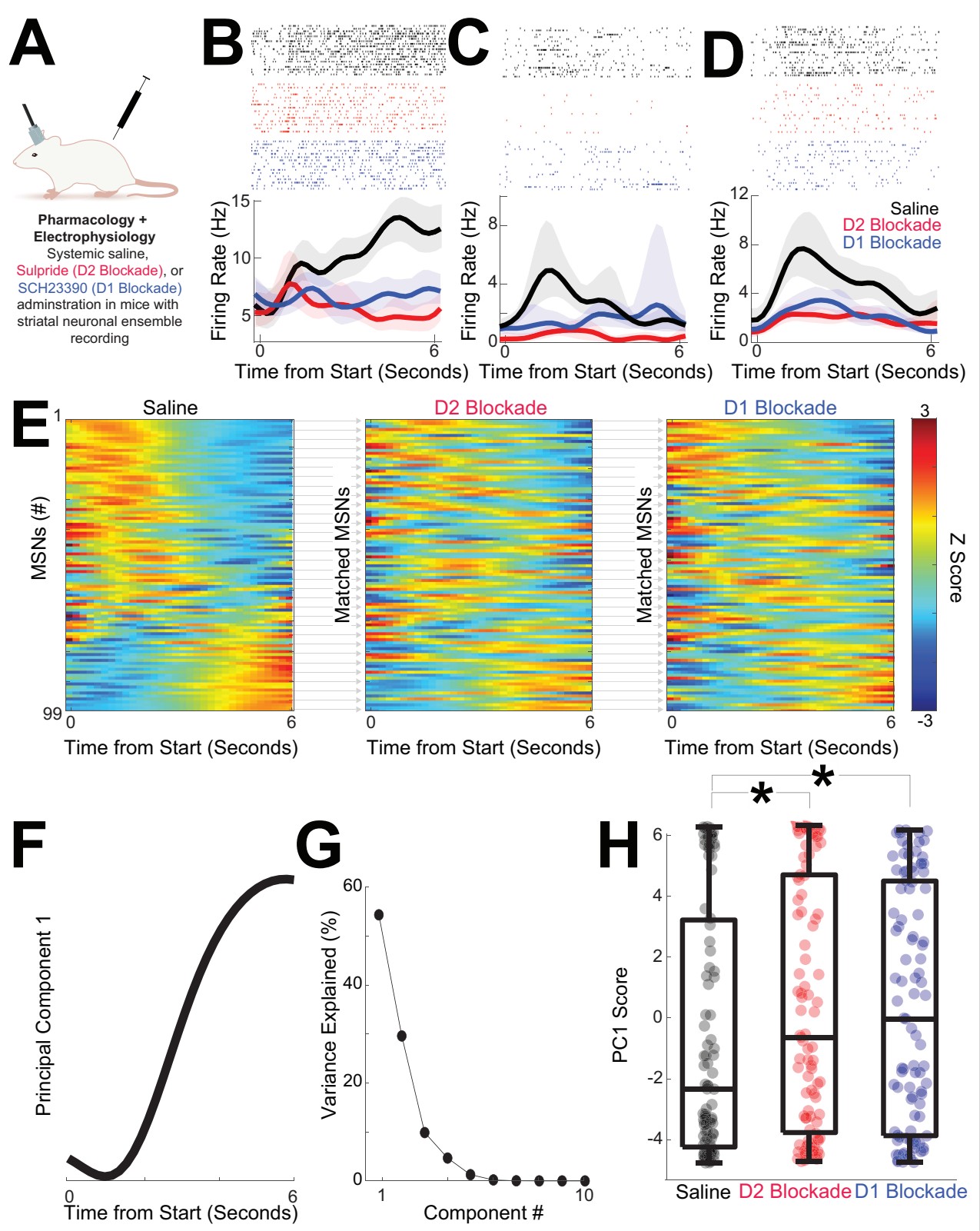

**Figure 6.** D2 and D1 blockade shift temporal dynamics. (**A**) We recorded dorsomedial striatal medium spiny neuron (MSN) ensembles during interval timing in sessions with saline, D2 blockade with sulpiride, or D1 blockade with SCH23390. (**B–D**) Example peri-event raster from MSNs in sessions with saline (black), D2-dopamine blockade (red), or D1-dopamine blockade (blue). Shaded area is the bootstrapped 95% confidence interval. (**E**) MSNs from 99 neurons in 11 mice from saline, D2 blockade, or D1 blockade session; MSNs were matched across sessions based on waveforms and interspike

*Figure 6 continued on next page*

*Figure 6 continued*

interval. Each row represents a peri-event time histogram (PETH) binned at 0.2 s, smoothed using kernel-density estimates using a bandwidth of 1, and z-scored. Colors indicate z-scored firing rate. See *Figure 6—figure supplement 1* for analyses that assume statistical independence. (**F**) Principal component analysis (PCA) identified MSN ensemble patterns of activity. The first principal component (PC1) exhibited time-dependent ramping. (**G**) PC1 explained 54% of population variance among MSN ensembles; higher components were not analyzed. (**H**) PC1 scores were closer to zero and significantly different with D2 or D1 blockade; *p < 0.05 via linear mixed effects; data from 99 MSNs in 11 mice.

The online version of this article includes the following figure supplement(s) for figure 6:

**Figure supplement 1.** We analyzed medium spiny neuron (MSN) ensembles in sessions with saline (158 neurons), D2 blockade (167 neurons), or D1 blockade (144 neurons) – unlike *Figure 6*; all sessions were sorted independently and assumed to be fully statistically independent.

**Figure supplement 2.** PC1 scores from individual mice in *Figure 6H*.

**Figure supplement 3.** MSN classification by waveform criteria for pharmacology sessions.

between D2 and D1 blockades. Furthermore, PC1 was distinct even when sessions were sorted independently and assumed to be fully statistically independent (*Figure 6—figure supplement 1*). Higher components explained less variance and were not reliably different between saline and D2 or D1 blockade. Taken together, this data-driven analysis shows that D2 and D1 blockades produced similar shifts in MSN population dynamics represented by PC1. When combined with the major contributions of D1/D2 MSNs to PC1 (*Figure 3C*) these findings indicate that pharmacological D2 and D1 blockades disrupt ramping-related activity in the striatum.

### D2 and D1 blockades degrade MSN temporal decoding

Finally, we quantified striatal MSN temporal decoding via a naive Bayesian classifier that generates trial-by-trial predictions of time from MSN ensemble firing rates (*Figure 7A–C*; *Bruce et al., 2021*; *Emmons et al., 2017*). Our DDMs predict that disrupted temporal decoding would be a consequence of an altered DDM drift rate. We used leave-one-out cross-validation to predict objective time from the firing rate within a trial. Saline sessions generated strong temporal predictions for the first 6 s of the interval immediately after trial start (0–6 s; $R^2$ = 0.91 (0.83–0.94)) with weaker predictions for later epochs (6–12 s: $R^2$ = 0.55 (0.34–0.70); rank sum p = 0. 000002 vs 0–6 s, Cohen $d$ = 2.0; 12–18 s: $R^2$ = 0.18 (0.10–0.62); rank sum p = 0.000003 vs 0–6 s, Cohen $d$ = 2.4; all analyses considered each epoch statistically independent; *Figure 7D*). We found that temporal decoding early in the interval (0–6 s) was degraded with either D2 blockade ($R^2$ = 0.69 (0.58–0.84); rank sum p = 0.0002 vs saline, Cohen $d$ = 1.4) or D1 blockade ($R^2$ = 0.71 (0.47–0.87); rank sum p = 0.004 vs saline, Cohen $d$ = 1.1; *Figure 7D*), consistent with predictions made from DDMs. Later in the interval (6–12 and 12–18 s), there were no significant differences between saline sessions and D2 or D1 blockade (*Figure 7D*).

Taken together, these data demonstrate that disrupting either D2-MSNs or D1-MSNs degrades temporal decoding by MSN ensembles. In combination with our optogenetic tagging, computational modeling, and optogenetic inhibition experiments, these data provide insight into cognitive computations by the striatum.

## Discussion

We describe how striatal MSNs work together in complementary ways to encode an elementary cognitive process, interval timing. Strikingly, optogenetic tagging showed that D2-MSNs and D1-MSNs had distinct dynamics during interval timing. MSN dynamics helped construct and constrain a four-parameter DDM in which D2- and D1-MSN spiking accumulated temporal evidence. This model predicted that disrupting either D2-MSNs or D1-MSNs would increase switch response times. Accordingly, we found that optogenetically or pharmacologically disrupting striatal D2-MSNs or D1-MSNs increased switch response times without affecting task-specific movements. Disrupting D2-MSNs or D1-MSNs shifted MSN temporal dynamics and degraded MSN temporal decoding. These data, when combined with our model predictions, demonstrate that despite opposing dynamics, D2-MSNs and D1-MSNs contribute complementary temporal evidence to controlling actions in time. Our interpretation is that because the activity of D2-MSN and D1-MSN ensembles represents the accumulation evidence, pharmacological/optogenetic disruption of D2-MSN/D1-MSN activity slows this accumulation process, leading to slower interval timing-response times (*Figure 5*) without changing other task-specific movements (*Figure 5—figure supplement 3*). These results provide new insight into how

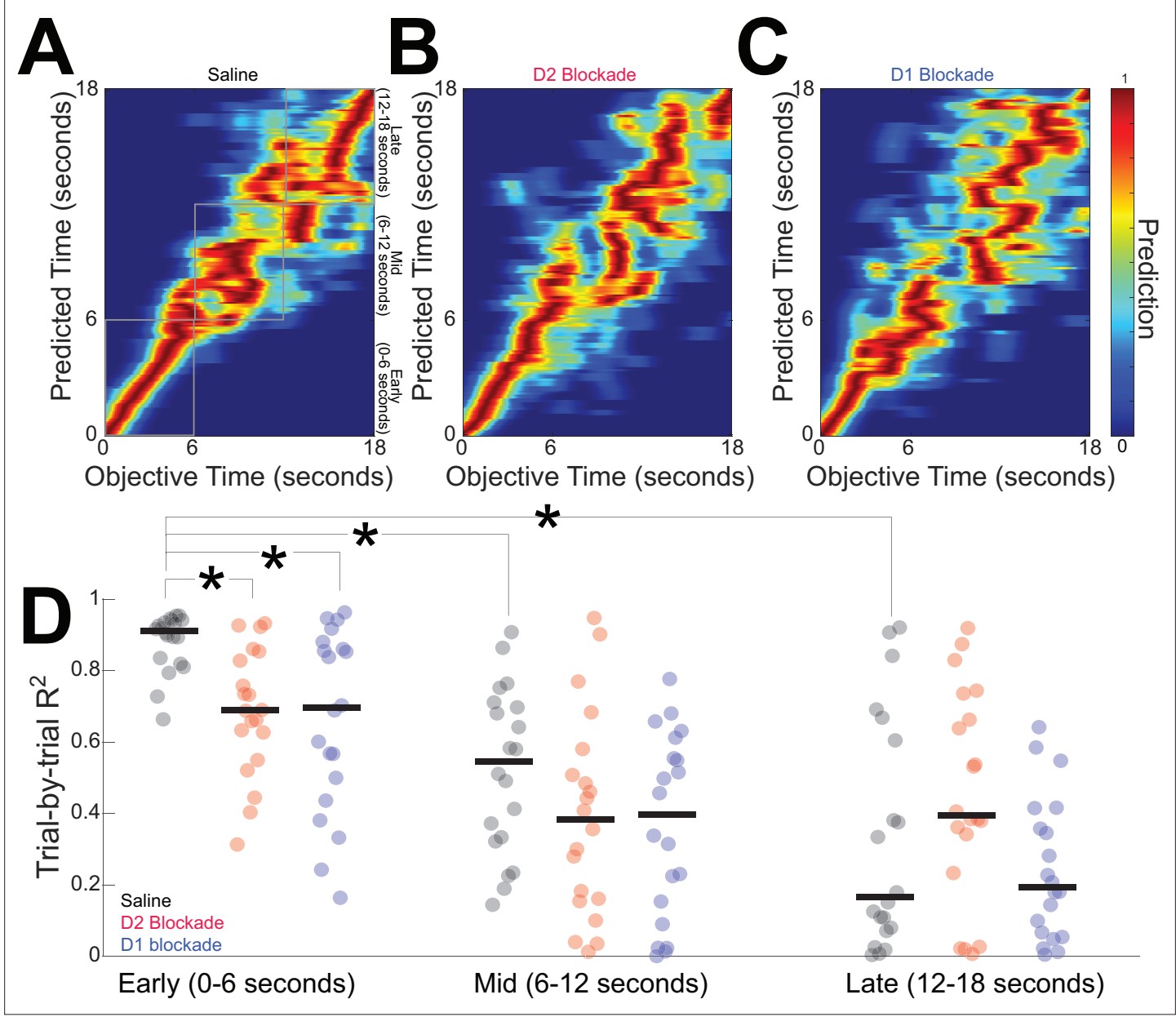

**Figure 7.** D2 and D1 blockades degrade medium spiny neuron (MSN) temporal decoding. We used naive Bayesian classifiers to decode time from MSN ensembles in (**A**) saline sessions, (**B**) D2 blockade sessions, and (**C**) D1 blockade sessions. Color represents the temporal prediction across 20 trials with red representing stronger predictions. (**D**) Temporal decoding was strong early in the interval, and D2 or D1 blockade degraded classification accuracy. Temporal decoding was decreased later in the interval. Each point represents the $R^2$ for each trial of behavior for MSN ensembles from 11 mice. *$p <$ 0.05 vs saline from 0 to 6 s. Horizontal black lines in (**D**) represent group medians.

opposing patterns of striatal MSN activity control behavior in similar ways and show that they play a complementary role in elementary cognitive operations.

Striatal MSNs are critical for temporal control of action (*Emmons et al., 2017*; *Gouvêa et al., 2015*; *Mello et al., 2015*). Three broad models have been proposed for how striatal MSN ensembles represent time: (1) the striatal beat frequency model, in which MSNs encode temporal information based on neuronal synchrony *Matell and Meck, 2004*; (2) the distributed coding model, in which time is represented by the state of the network *Paton and Buonomano, 2018*; and (3) the DDM, in which neuronal activity monotonically drifts toward a threshold after which responses are initiated (*Emmons et al., 2017*; *Simen et al., 2011*; *Wang et al., 2018*). While our data do not formally resolve these possibilities, our results show that D2-MSNs and D1-MSNs exhibit opposing changes in firing rate

dynamics in PC1 over the interval. Past work by our group and others has demonstrated that PC1 dynamics can scale over multiple intervals to represent time (*Emmons et al., 2020*; *Emmons et al., 2017*; *Gouvêa et al., 2015*; *Mello et al., 2015*; *Wang et al., 2018*). We find that low parameter DDMs account for interval timing behavior with both intact and disrupted striatal D2- and D1-MSNs. While other models can capture interval timing behavior and account for MSN neuronal activity, our model does so parsimoniously with relatively few parameters (*Matell and Meck, 2004*; *Paton and Buonomano, 2018*; *Simen et al., 2011*). We and others have shown previously that ramping activity scales to multiple intervals, and DDMs can be readily adapted by changing the drift rate (*Emmons et al., 2017*; *Gouvêa et al., 2015*; *Mello et al., 2015*; *Simen et al., 2011*). Interestingly, decoding performance was high early in the interval; indeed, animals may have been focused on this initial interval (*Balci and Gallistel, 2006*) in making temporal comparisons and deciding whether to switch response nosepokes.

D2-MSNs and D1-MSNs play complementary roles in movement. For instance, stimulating D1-MSNs facilitates movement, whereas stimulating D2-MSNs impairs movement (*Kravitz et al., 2010*). Both populations have been shown to have complementary patterns of activity during movements with MSNs firing at different phases of action initiation and selection (*Tecuapetla et al., 2016*). Further dissection of action selection programs reveals that opposing patterns of activation among D2-MSNs and D1-MSNs suppress and guide actions, respectively, in the dorsolateral striatum (*Cruz et al., 2022*). A particular advantage of interval timing is that it captures a cognitive behavior within a single dimension — time. When projected along the temporal dimension, it was surprising that D2-MSNs and D1-MSNs had opposing patterns of activity. Ramping activity in the prefrontal cortex can increase or decrease and prefrontal neurons project to and control striatal ramping activity (*Emmons et al., 2020*; *Emmons et al., 2017*; *Wang et al., 2018*). It is possible that differences in D2-MSNs and D1-MSNs reflect differences in cortical ramping, which may themselves reflect more complex integrative or accumulatory processes. Further experiments are required to investigate these differences. Past pharmacological work from our group and others has shown that disrupting D2- or D1-MSNs slows timing (*De Corte et al., 2019*; *Drew et al., 2007*; *Drew et al., 2003*; *Stutt et al., 2024*) and are in agreement with pharmacological and optogenetic results in this manuscript. Computational modeling predicted that disrupting either D2-MSNs or D1-MSNs increased self-reported estimates of time, which was supported by both optogenetic and pharmacological experiments.

Notably, these disruptions are distinct from increased timing variability reported with administrations of amphetamine, ventral tegmental area dopamine neuron lesions, and rodent models of aging (*Balci et al., 2008*; *Gür et al., 2020*; *Weber et al., 2024b*; *Weber et al., 2023*). Furthermore, our current data demonstrate that disrupting either D2-MSN or D1-MSN activity shifted MSN dynamics and degraded temporal decoding, supporting prior work (*De Corte et al., 2019*; *Drew et al., 2007*; *Drew et al., 2003*; *Stutt et al., 2024*). Our recording experiments do not identify where a possible response threshold *T* is instantiated, but downstream basal ganglia structures may have a key role in setting response thresholds (*Toda et al., 2017*).

Because interval timing is reliably disrupted in human diseases of the striatum such as Huntington's disease, Parkinson's disease, and schizophrenia (*Hinton et al., 2007*; *Singh et al., 2021*; *Ward et al., 2012*), these results have relevance to human disease. Our task version has been used extensively to study interval timing in mice and humans (*Balci et al., 2008*; *Bruce et al., 2021*; *Stutt et al., 2024*; *Tosun et al., 2016*; *Weber et al., 2023*). However, temporal bisection tasks, in which animals hold during a temporal cue and respond at different locations depending on cue length, have advantages in studying how animals time an interval because animals are not moving while estimating cue duration (*Paton and Buonomano, 2018*; *Robbe, 2023*; *Soares et al., 2016*). Our interval timing task version – in which mice switch between two response nosepokes to indicate their interval estimate has elapsed – has been used extensively in rodent models of neurodegenerative disease (*Larson et al., 2022*; *Weber et al., 2024a*; *Weber et al., 2023*; *Zhang et al., 2021*), as well as in humans (*Stutt et al., 2024*). This version of interval timing involves temporal control of movements, which engages executive function and has more translational relevance for human diseases than perceptual timing or bisection tasks (*Brown, 2006*; *Farajzadeh and Sanayei, 2024*; *Zhang et al., 2016*; *Singh et al., 2021*). Many therapeutics targeting dopamine receptors are used clinically, and our findings help describe how dopaminergic drugs might affect striatal function and dysfunction. Future studies of D2-MSNs and D1-MSNs in temporal bisection and

other timing tasks may further clarify the relative roles of D2- and D1-MSNs in interval timing and time estimation.

Our approach has several limitations. First, systemic drug injections block D2 and D1 receptors in many different brain regions, including the frontal cortex, which is involved in interval timing (*Kim et al., 2017*). Our past work locally infused drugs blocking D2- and D1-dopamine receptors within the dorsomedial striatum, and found increased response times during interval timing as in *Figure 5* (*De Corte et al., 2019*). Pharmacological interventions may have complex effects that might affect D2-MSN or D1-MSN ensembles. We note that optogenetic inhibition of D2-MSNs and D1-MSNs produces similar effects to pharmacology in *Figure 5*. Pharmacology is compatible with neuronal ensemble recordings as optogenetic inhibition would silence a large fraction of neurons, complicating interpretations. Future studies might extend our work by combining local pharmacology with neuronal ensemble recording. Second, although we had adequate statistical power and medium-to-large effect sizes, optogenetic tagging is low yield, and it is possible that recording more of these neurons would afford greater opportunity to identify more robust results and alternative coding schemes, such as neuronal synchrony. Recording more neurons simultaneously would also help further constrain DDMs to predict trial-by-trial firing rate and generate a more sophisticated neuronal network model of time (*Figure 4—figure supplement 1*). There was some variability within animals (*Figure 3—figure supplement 1* and *Figure 6—figure supplement 2*), although some of this variability was explained by firing rate slope and whether MSNs ramped or down (*Figure 3C, D*). Third, the striatum includes diverse cell types, some of which express both D1- and D2-dopamine receptors; these cell types may also contribute to cognitive processing. Fourth, MSNs can laterally inhibit each other, which may profoundly affect striatal function. Regardless, we show that cell-type-specific disruption of D2-MSNs and D1-MSNs both slow timing, implying that these cell types play a key role in interval timing. Future experiments may record from non-MSN striatal cell types, including fast-spiking interneurons that shape basal ganglia output. Fifth, we did not deliver stimulation to the striatum because our pilot experiments triggered movement artifacts or task-specific dyskinesias (*Kravitz et al., 2010*). Stimulation approaches carefully titrated to striatal physiology may affect interval timing without affecting movement. Finally, movement and motivation can contribute to MSN dynamics (*Robbe, 2023*). Four lines of evidence argue that our findings cannot be directly explained by motor confounds: (1) D2-MSNs and D1-MSNs diverge between 0 and 6 s after trial start well before the first nosepoke (*Figure 2—figure supplement 2*), (2) our GLM accounted for nosepokes and nosepoke-related βs were similar between D2-MSNs and D1-MSNs, (3) optogenetic disruption of dorsomedial D2-MSNs and D1-MSNs did not change task-specific movements despite reliable changes in switch response time, and (4) ramping dynamics were quite distinct from movement dynamics. Furthermore, disrupting D2-MSNs and D1-MSNs did not change the number of rewards animals received, implying that these disruptions did not grossly affect motivation. Still, future work combining motion tracking/accelerometry with neuronal ensemble recording and optogenetics and including bisection tasks may further unravel timing vs movement in MSN dynamics (*Robbe, 2023*; *Tecuapetla et al., 2016*).

In summary, we examined the role of dorsomedial striatal D2-MSNs and D1-MSNs during an elementary cognitive behavior, interval timing. Optogenetic tagging revealed that D2-MSNs and D1-MSNs exhibited opposite and complementary patterns of neuronal activity. These dynamics could be captured by computational DDMs, which predicted that disrupting either D2-MSNs or D1-MSNs would slow the accumulation of temporal evidence and increase switch response time. In concordance with this prediction, we found that optogenetic or pharmacological disruption of either D2-MSNs or D1-MSNs increased switch response times, with pharmacological D2 or D1 blockade shifting MSN dynamics and degrading temporal decoding. Collectively, our data provide insight into how the striatum encodes cognitive information, which could be highly relevant for human diseases that disrupt the striatum and for next-generation neuromodulation that targets the basal ganglia to avoid cognitive side effects or to treat cognitive dysfunction.

## Materials and methods
### Rodents
All procedures were approved by the Institutional Animal Care and Use Committee (IACUC) at the University of Iowa, and all experimental methods were performed in accordance with applicable

guidelines and regulations (Protocol #0062039). We used five cohorts of mice, summarized in *Table 1*: (1) 30 wild-type C57BL/6J mice (17 female) for behavioral experiments (*Figure 1*); (2) 4 *Drd2-cre+* mice derived from Gensat strain ER44 (2 female) and 5 *Drd1-cre+* mice derived from Gensat strain EY262 (2 female) for optogenetic tagging and neuronal ensemble recordings in *Figures 2 and 3*; (3) 10 *Drd2-cre+* (5 female), and 6 *Drd1-cre+* mice (3 female) for optogenetic inhibition (*Figure 5*) with 5 *Drd2-cre+* and 5 *Drd1-cre+* controls; (4) 10 wild-type mice for behavioral pharmacology (*Figure 5*); and (5) 11 mice (4 C57BL/6J (2 female), 5 *Drd2-cre+* mice (2 female), and 2 *Drd1-cre+* mice (0 female)) for combined behavioral pharmacology and neuronal ensemble recording (*Figures 6 and 7*; *Table 1*). Our recent work shows that D2 and D1 blockades have similar effects in both sexes (*Stutt et al., 2024*).

## Interval timing switch task

We used a mouse-optimized operant interval timing task described in detail previously (*Balci et al., 2008*; *Bruce et al., 2021*; *Tosun et al., 2016*; *Weber et al., 2023*). Briefly, mice were trained in sound-attenuating operant chambers, with two front nosepokes flanking either side of a food hopper on the front wall, and a third nosepoke located at the center of the back wall. The chamber was positioned below an 8 kHz, 72 dB speaker (*Figure 1A*; MedAssociates, St. Albans, VT). Mice were 85% food restricted and motivated with 20 mg sucrose pellets (Bio-Serv, Flemington, NJ). Mice were initially trained to receive rewards during fixed ratio nosepoke response trials. Nosepoke entry and exit were captured by infrared beams. After shaping, mice were trained in the 'switch' interval timing task. Mice self-initiated trials at the back nosepoke, after which tone and nosepoke lights were illuminated simultaneously. Cues were identical on all trial types and lasted the entire duration of the trial (6 or 18 s). On 50% of trials, mice were rewarded for a nosepoke after 6 s at the designated first 'front' nosepoke; these trials were not analyzed. On the remaining 50% of trials, mice were rewarded for nosepoking first at the 'first' nosepoke location and then switching to the 'second' nosepoke location; the reward was delivered for initial nosepokes at the second nosepoke location after 18 s when preceded by a nosepoke at the first nosepoke location. Multiple nosepokes at each nosepoke were allowed. Early responses at the first or second nosepoke were not reinforced. Initial responses at the second nosepoke rather than the first nosepoke, alternating between nosepokes, going back to the first nosepoke after the second nosepoke were rare after initial training. Error trials included trials where animals responded only at the first or second nosepoke and were also not reinforced. We did not analyze error trials as they were often too few to analyze; these were analyzed at length in our prior work (*Bruce et al., 2021*).

*Switch response time* was defined as the moment animals departed the first nosepoke before arriving at the second nosepoke. Critically, switch responses are a time-based decision guided by temporal control of action because mice switch nosepokes only if nosepokes at the first location did not receive a reward after 6 s. That is, mice estimate if more than 6 s have elapsed without receiving a reward to decide to switch responses. Mice learn this task quickly (3–4 weeks), and error trials in which an animal nosepokes in the wrong order or does not nosepoke are relatively rare and discarded. Consequently, we focused on these switch response times as the key metric for temporal control of action. *Traversal time* was defined as the duration between first nosepoke exit and second nosepoke entry and is distinct from switch response time when animals departed the first nosepoke. *Nosepoke duration* was defined as the time between first nosepoke entry and exit for the switch response times only. Trials were self-initiated, but there was an intertrial interval with a geometric mean of 30 s between trials.

## Surgical and histological procedures

Surgical procedures were identical to methods described previously (*Bruce et al., 2021*). Briefly, mice were anesthetized using inhaled 4% isoflurane and surgical levels of anesthesia were maintained at 1–2% for the duration of the surgery. Craniotomies were drilled above bilateral dorsal striatal anatomical targets, and optogenetic viruses (AAV5-DIO-eNHPR2.0 (halorhodopsin), AAV5-DIO-ChR2(H134R)-mcherry (ChR2), or AAV5-DIO-cherry (control) from the University of North Carolina Viral Vector Core) were injected into the dorsal striatum (0.5 µl of virus, +0.9, ML +/–1.3, DV –2.7). Either fiber optics (Doric Lenses, Montreal Quebec; AP +0.9, ML +/–1.3, DV –2.5) or 4 × 4 electrode or optrode arrays (AP +0.4, ML –1.4, DV –2.7 on the left side only; Microprobes, Gaithersburg, MD) were positioned

in the dorsal striatum. Holes were drilled to insert skull screws to anchor headcap assemblies and/or ground electrode arrays and then sealed with cyanoacrylate ('SloZap', Pacer Technologies, Rancho Cucamonga, CA), accelerated by 'ZipKicker' (Pacer Technologies) and methyl methacrylate (AM Systems, Port Angeles, WA). Following postoperative recovery, mice were trained on the switch task and acclimated to experimental procedures prior to undergoing experimental sessions.

## Optogenetics

We leveraged cell-type-specific optogenetics to manipulate D2-MSNs and D1-MSNs in D2- or D1-cre mice. In animals injected with optogenetic viruses, optical inhibition was delivered via bilateral patch cables for the entire trial duration of 18 s via 589 nm laser light at 12 mW power on 50% of randomly assigned trials. To control for heating and nonspecific effects of optogenetics, we performed control experiments in mice without opsins using identical laser parameters in D2- or D1-cre mice (*Figure 5— figure supplement 2*). We did not stimulate for epochs less than the interval because we did not want to introduce a cue during the interval. For optogenetic tagging, putative D1- and D2-MSNs were optically identified via 473 nm photostimulation (*Figure 2—figure supplement 1*). Units with mean post-stimulation spike latencies of ≤5 ms and a stimulated-to-unstimulated waveform correlation ratio of >0.9 were classified as putative D2-MSNs or D1-MSNs (*Ryan et al., 2018*; *Shin et al., 2018*). Only one recording session was performed for each animal per day, and one recording session was included from each animal.

## Behavioral pharmacology procedures

C57BL/6J mice were injected IP 20–40 min before interval timing trials with either SCH23390 ($C_{17}H_{18}ClNO$; D1 blockade), sulpiride ($C_{15}H_{23}N_3O_4S$; D2 blockade), or isotonic saline. The sulpiride dosage was 12.5 mg/kg, 0.01 ml/g, and the SCH23390 was administered at a dosage of 0.05 mg/kg, 0.01 ml/g. Behavioral performance was compared with interval timing behavior on the prior day when isotonic saline was injected IP. Only one recording session was performed for each animal per day, and one recording session was included from saline, D2 blockade, and D1 blockade sessions.

## Electrophysiology

Single-unit recordings were made using a multi-electrode recording system (Open Ephys, Atlanta, GA). After the experiments, Plexon Offline Sorter (Plexon, Dallas, TX), was used to remove artifacts. PCA and waveform shape were used for spike sorting. Single units were defined as those (1) having a consistent waveform shape, (2) being a separable cluster in PCA space, and (3) having a consistent refractory period of at least 2 ms in interspike interval histograms. The same MSNs were sorted across saline, D2 blockade, and D1 blockade sessions by loading all sessions simultaneously in Offline Sorter and sorted using the preceding criteria. MSNs had to have consistent firing in all sessions to be included. Sorting integrity across sessions was quantified by comparing waveform similarity via correlation coefficients between sessions.

Spike activity was analyzed for all cells that fired between 0.5 and 20 Hz over the entire behavioral session. Putative MSNs were further separated from striatal fast-spiking interneurons based on hierarchical clustering of the waveform peak-to-trough ratio and the half-peak width (*fitgmdist* and *cluster.m*; *Figure 2—figure supplement 1*; *Berke, 2011*). We calculated kernel density estimates of firing rates across the interval (–4 s before trial start to 22 s after trial start) binned at 0.2 s, with a bandwidth of 1. We used PCA to identify data-driven patterns of *z*-scored neuronal activity, as in our past work (*Bruce et al., 2021*; *Emmons et al., 2017*; *Kim et al., 2017*). The variance of PC1 was empirically compared against data generated from 1000 iterations of data from random timestamps with identical bins and kernel density estimates. Average plots were shown with Gaussian smoothing for plotting purposes only.

## Immunohistochemistry

Following completion of experiments, mice were transcardially perfused with ice-cold 1× phosphate-buffered saline and 4% paraformaldehyde (PFA) after anesthesia using ketamine (100 mg/kg IP) and xylazine (10 mg/kg IP). Brains were then fixed in solutions of 4% PFA and 30% sucrose before being cryosectioned on a freezing microtome. Sections were stained for tyrosine hydroxylase with primary antibodies for >12 hr (rabbit anti-TH; Millipore MAB152; 1:1000) at 4°C. Sections were subsequently

visualized with Alexa Fluor fluorescent secondary antibodies (goat anti-rabbit IgG Alexa 519; Thermo Fisher Scientific; 1:1000) matched to host primary by incubating 2 hr at room temperature. Histological reconstruction was completed using postmortem analysis of electrode placement by slide-scanning microscopy on an Olympus VS120 microscope (Olympus, Center Valley, PA).

### Trial-by-trial GLMs

To measure time-related ramping over the first 6 s of the interval, we used trial-by-trial GLMs at the individual neuron level in which the response variable was firing rate and the predictor variable was time in the interval or nosepoke rate (*Shimazaki and Shinomoto, 2007*). For each neuron, its time-related 'ramping' slope was derived from the GLM fit of firing rate vs time in the interval, for all trials per neuron. GLMs accounted for nosepoke movements unless there were no nosepokes within the 0- to 6-s interval. All GLMs were run at a trial-by-trial level to avoid the effects of trial averaging (*Latimer et al., 2015*) as in our past work (*Bruce et al., 2021*; *Emmons et al., 2017*). We performed additional sensitivity analyses excluding outliers outside of 95% confidence intervals and measuring the firing rate from the start of the interval to the time of the switch response on a trial-by-trial level for each neuron.

### Machine-learning analyses

To predict time, we used a naive Bayesian classifier to evaluate neuronal ensemble decoding as in our past work (*Emmons et al., 2020*; *Emmons et al., 2017*; *Gouvêa et al., 2015*; *Kim et al., 2017*; *Mello et al., 2015*). Data from neurons with more than 20 trials across all mice with the goal of evaluating how the decoding of time is affected by the epoch in the interval and by D2/D1 blockade. To preclude edge effects that might bias classifier performance, we included data from 6 s before trial start and 6 s after the end of the interval. We used leave-one-out cross-validation to predict an objective time from the firing rate within a trial. Classifier performance was quantified by computing the $R^2$ of objective time vs predicted time, only for bins during the interval (0–6, 6–12, and 12–18 s; see *Figure 7*). Classifier performance was compared using time-shuffled firing rates via a Wilcoxon signed-rank test.

### Drift-diffusion models

We constructed a four-parameter DDM (see *Equations 1–3* in Results) that accounted for the ensemble threshold behavior observed in the neural data, then used it to simulate switch response times compatible with mice behavioral data. The model was implemented in MATLAB, starting from initial value $b$ and with discrete computer simulation steps $x_{new} = x_{old} + (F - x_{old})D\Delta t + \sigma\sqrt{\Delta t}N(0,1)$. Here $N(0,1)$ represents random values generated from the standardized Gaussian distribution (mean = 0 and standard deviation = 1). The integration timestep was $\Delta t = 0.1$. For each numerical simulation, the 'switch response time' was defined as the time $t^*$ when variable $x$ first reached the threshold value $T = F(1 - b/4) + (1 - F)b/4$ For each condition, we ran 500 simulations of the model (up to 25 s per trial) and recorded the switch response times. Examples of firing rate $x$ dynamics are shown in *Figure 4A, B*. We observed that the distributions of interval timing switch response times could be fit by a gamma probability distribution function $PDF(x) = \frac{\beta^\alpha}{\Gamma(\alpha)}x^{\alpha-1}e^{-\beta x}$ with shape $\alpha$ and rate $\beta$ (see *Figure 4—figure supplement 3* and *Supplementary file 1*). Tenfold cross-validation revealed highly stable fits between gamma, models, and data.

### Selection of DDM parameters

Our goal was to build DDMs with dynamics that produce 'response times' according to the observed distribution of mice switch times. The selection of parameter values in *Figure 4* was done in three steps. First, we fit the distribution of the mice behavioral data with a Gamma distribution and found its fitting values for shape $\alpha_M$ and rate $\beta_M$ (*Supplementary file 1* and *Figure 4—figure supplement 3*; $R^2$ Data vs Gamma ≥0.94). We recognized that the mean $\mu_M$ and the coefficient of variation $CV_M$ are directly related to the shape and rate of the Gamma distribution by formulas $\mu_M = \alpha_M/\beta_M$ and $CV_M = 1/\sqrt{\alpha_M}$. Next, we fixed parameters $F$ and $b$ in DDM (e.g., for D2-MSNs: $F = 1$, $b = 0.52$) and simulated the DDM for a range of values for $D$ and $\sigma$. For each pair $(D, \sigma)$, one computational 'experiment' generated 500 response times with mean μ and coefficient of variation $CV$. We repeated the 'experiment' 10 times and took the group median of μ and $CV$ to obtain the simulation-based statistical measures $\mu_S$ and $CV_S$. Last, we plotted $E^\mu = |(\mu_S - \mu_M)/\mu_M|$ and $E^{cv} = |CV_S - CV_M|$, the respective

relative error and the absolute error to data (*Figure 4—figure supplement 2*). We considered that parameter values $(D, \sigma)$ provided a good DDM fit of mice behavioral data whenever $E^{\mu} \leq 0.05$ and $E^{cv} \leq 0.02$ (*Figure 4—figure supplement 2*).

## Analysis and modeling of mouse MSN ensemble recordings

Our preliminary analysis found that, for sufficiently large number of neurons ($N > 11$), each recorded ensemble of MSNs on a trial-by-trial basis could predict when mice would respond. We took the following approach: First, for each MSN, we convolved its trial-by-trial spike train $Spk(t)$ with a 1-s exponential kernel $K(t) = we^{-t/w}$ if $t > 0$ and $K(t) = 0$ if $t \leq 0$ (here w=1 s). Therefore, the smoothed, convolved spiking activity of neuron $j$ ($j = 1, 2, \ldots N$),

$$x_j(t) = \left(Spk_j * K\right)(t) = \int_{-\infty}^{\infty} K(s)Spk_j\left(t - s\right) ds = \int_{0}^{\infty} K(s)Spk_j\left(t - s\right) ds$$

tracks and accumulates the most recent (one second, in average) firing-rate history of the $j$ th MSN, up to moment $t$. We hypothesized that the ensemble activity $(x_1(t), x_2(t), \ldots, x_N(t))$, weighted with some weights $\beta_j$, could predict the trial switch time $t^*$ by considering the sum

$$x(t) = \sum_{j=1}^{N} \beta_j x_j(t)$$

and the sigmoid

$$y(t) = \frac{1}{1 + \exp\left(-(x(t) - 0.5)/t\right)} = \frac{1}{1 + \exp\left[-\left(\hat{\beta}_0 + \sum_{j=1}^{N} \hat{\beta}_j x_j(t)\right)\right]}$$

that approximates the firing rate of an output unit. Here parameter $k$ indicates how fast $x(t)$ crosses the threshold 0.5 coming from below (if $k > 0$) or coming from above (if $k < 0$) and relates the weights $\beta_j$ to the unknowns $\hat{\beta}_j = \beta_j/k$ and $\hat{\beta}_0 = -0.5/k$. Next, we ran a logistic fit for every trial for a given mouse over the spike count predictor matrix $(x_1(t), x_2(t), \ldots, x_N(t))$ from the mouse MSN recorded ensemble, and observed value $t^*$, estimating the coefficients $\hat{\beta}_0$ and $\beta_j$, and so, implicitly, the weights $\beta_j$. From there, we compute the predicted switch time $t^*_{pred}$ by condition $x(t) = 0.5$. Accuracy was quantified comparing the predicted accuracy within a 1-s window to switch time on a trial-by-trial basis (*Figure 4—figure supplement 1*).

## Statistics

All data and statistical approaches were reviewed by the Biostatistics, Epidemiology, and Research Design Core (BERD) at the Institute for Clinical and Translational Sciences (ICTS) at the University of Iowa. All code and data are made available at https://narayanan.lab.uiowa.edu/article/datasets. We used the median to measure central tendency and the interquartile range to measure spread. We used Wilcoxon nonparametric tests to compare behavior between experimental conditions and Cohen's $d$ to calculate effect size. Analyses of putative single-unit activity and basic physiological properties were carried out using custom routines for MATLAB.

For all neuronal analyses, variability between animals was accounted for using generalized linear mixed-effects models and incorporating a random effect for each mouse into the model, which allows us to account for inherent between-mouse variability. We used *fitglme* in MATLAB and verified main effects using *lmer* in R. We accounted for variability between MSNs in pharmacological datasets in which we could match MSNs between saline, D2 blockade, and D1 blockade. p-values <0.05 were interpreted as significant.

## Acknowledgements

This work was funded by NIMH R01MH116043 to NSN. We thank editors, reviewers, Bernardo Sabatini, Filipe Rodrigues, and Joseph Paton for detailed engagement with our manuscript.

# Additional information

## Funding

| Funder | Grant reference number | Author |
|---|---|---|
| National Institute of Mental Health | R01MH116043 | Nandakumar S Narayanan |

The funders had no role in study design, data collection, and interpretation, or the decision to submit the work for publication.

## Author contributions

Robert A Bruce, Conceptualization, Data curation, Formal analysis, Investigation, Methodology, Software, Writing – original draft, Writing – review and editing; Matthew Weber, Alexandra Bova, Conceptualization, Data curation, Formal analysis, Investigation, Software, Writing – original draft, Writing – review and editing; Rachael Volkman, Methodology, Writing – original draft, Writing – review and editing; Casey Jacobs, Methodology; Kartik Sivakumar, Hannah Stutt, Youngcho Kim, Methodology, Writing – review and editing; Rodica Curtu, Methodology, Writing – original draft, Project administration, Writing – review and editing; Nandakumar S Narayanan, Conceptualization, Data curation, Formal analysis, Investigation, Project administration, Software, Supervision, Writing – original draft, Writing – review and editing

## Author ORCIDs

Rodica Curtu ⓘ https://orcid.org/0000-0003-2163-4689
Nandakumar S Narayanan ⓘ https://orcid.org/0000-0002-0427-0003

## Ethics

All procedures were approved by the Institutional Animal Care and Use Committee (IACUC) at the University of Iowa, and all experimental methods were performed in accordance with applicable guidelines and regulations (Protocol #0062039).

Reviewer #1 (Public review): https://doi.org/10.7554/eLife.96287.4.sa1
Reviewer #2 (Public review): https://doi.org/10.7554/eLife.96287.4.sa2
Reviewer #3 (Public review): https://doi.org/10.7554/eLife.96287.4.sa3
Author response https://doi.org/10.7554/eLife.96287.4.sa4

# Additional files

## Supplementary files

Supplementary file 1. Gamma distribution parameters.
MDAR checklist

## Data availability

All raw data are available on Dryad at https://doi.org/10.5061/dryad.g4f4qrg15. All code is available at https://github.com/nandakumar-narayanan/https-elifesciences.org-reviewed-preprints-96287eLife/ (copy archived at *Narayanan, 2024*).

The following dataset was generated:

| Author(s) | Year | Dataset title | Dataset URL | Database and Identifier |
|---|---|---|---|---|
| Narayanan N | 2024 | Complementary cognitive roles for D2-MSNs and D1-MSNs during interval timing | https://doi.org/10.5061/dryad.g4f4qrg15 | Dryad Digital Repository, 10.5061/dryad.g4f4qrg15 |

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
