## [Editor Report · eLife Assessment]

This **valuable** study examines the activity and function of dorsomedial striatal neurons in the estimation of time. The authors examine striatal activity as a function of time as well as the impact of optogenetic striatal manipulation on the animal's ability to estimate a time interval, providing **solid** evidence for their claims. The study could be further strengthened with a more rigorous characterization of activity and a stronger connection between their proposed model and the experimental data. The work will be of interest to neuroscientists examining how striatum contributes to behavior.

---

## [Referee Report · Reviewer #1 (Public review)]

Summary:

In this work, the authors examine the activity and function of D1 and D2 MSNs in dorsomedial striatum (DMS) during an interval timing task. In this task, animals must first nosepoke into a cued port on the left or right; if not rewarded after 6 seconds, they must switch to the other port. Thus, this task requires animals to estimate if at least 6 seconds have passed after the first nosepoke. After verifying that animals estimate the passage of 6 seconds, the authors examine striatal activity during this interval. They report that D1-MSNs tend to decrease activity, while D2-MSNs increase activity, throughout this interval. They suggest that this activity follows a drift-diffusion model, in which activity increases (or decreases) to a threshold after which a decision is made. The authors next report that optogenetically inhibiting D1 or D2 MSNs, or pharmacologically blocking D1 and D2 receptors, increased the average wait time. This suggests that both D1 and D2 neurons contribute to the estimate of time, with a decrease in their activity corresponding to a decrease in the rate of 'drift' in their drift-diffusion model. Lastly, the authors examine MSN activity while pharmacologically inhibiting D1 or D2 receptors. The authors observe most recorded MSNs neurons decrease their activity over the interval, with the rate decreasing with D1/D2 receptor inhibition.

Major strengths:

The study employs a wide range of techniques - including animal behavioral training, electrophysiology, optogenetic manipulation, pharmacological manipulations, and computational modeling. The question posed by the authors - how striatal activity contributes to interval timing - is of importance to the field and has been the focus of many studies and labs. This paper contributes to that line of work by investigating whether D1 and D2 neurons have similar activity patterns during the timed interval, as might be expected based on prior work based on striatal manipulations. However, the authors find that D1 and D2 neurons have distinct activity patterns. They then provide a decision-making model that is consistent with all results. The data within the paper is presented very clearly, and the authors have done a nice job presenting the data in a transparent manner (e.g., showing individual cells and animals). Overall, the manuscript is relatively easy to read and clear, with sufficient detail given in most places regarding the experimental paradigm or analyses used.

Major weaknesses:

The results are based on a relatively small dataset (tens of cells).

Impact:

The task and data presented by the authors are very intriguing, and there are many groups interested in how striatal activity contributes to the neural perception of time. The authors perform a wide variety of experiments and analysis to examine how DMS activity influences time perception during an interval-timing task, allowing for insight into this process.

---

## [Referee Report · Reviewer #2 (Public review)]

This study found that D1-MSNs and D2-MSNs have opposing dynamics during interval timing in a mouse-optimized interval timing task. Further optogenetic and pharmacologic inhibition of either D1 or D2 MSNs increased response time. This study provides useful experimental evidence in the coding of time in striatum. However, there are some major weaknesses in this study.

(1) Regarding the data in Figure S3, The variance within each mouse was too big, the authors need to figure out and explain what caused the large variance within the same mouse, or the authors need to increase the sample size.

(2) Regarding the results in Figure 3 C and D, Figure 6 H and Figure 7 D, what is the sample size? From the single data points in the figures, it seems that the authors were using the number of cells to do statistical tests and plot the figures. For example, Figure 3 C, if the authors use n = 32 D2 MSNs and n = 41D1 MSNs to do the statistical test, it could make small difference to be statistically significant. The authors should use the number of mice to do the statistical tests.

(3) Regarding the results in Figure 5, what is the reason for the increase in the response times? The authors should plot the position track during intervals (0-6 s) with or without optogenetic or pharmacologic inhibition. The authors can check Figure 3, 5, and 6 in paper https://doi.org/10.1016/j.cell.2016.06.032 for reference to analyze the data.

---

## [Referee Report · Reviewer #3 (Public review)]

Summary:

The cognitive striatum, also known as the dorsomedial striatum, receives input from brain regions involved in high-level cognition and plays a crucial role in processing cognitive information. However, despite its importance, the extent to which different projection pathways of the striatum contribute to this information processing remains unclear. In this paper, Bruce et al. conducted a study using various causal and correlational techniques to investigate how these pathways collectively contribute to interval timing in mice. Their results were consistent with previous research, showing that the direct and indirect striatal pathways perform opposing roles in processing elapsed time. Based on their findings, the authors proposed a revised computational model in which two separate accumulators track evidence for elapsed time in opposing directions. These results have significant implications for understanding the neural mechanisms underlying cognitive impairment in neurological and psychiatric disorders, as disruptions in the balance between direct and indirect pathway activity are commonly observed in such conditions.

Strengths:

The authors employed a well-established approach to study interval timing and employed optogenetic tagging to observe the behavior of specific cell types in the striatum. Additionally, the authors utilized two complementary techniques to assess the impact of manipulating the activity of these pathways on behavior. Finally, the authors utilized their experimental findings to enhance the theoretical comprehension of interval timing using a computational model.

Weaknesses:

The behavioral task used in this study is best suited for investigating elapsed time perception rather than interval timing. Timing bisection tasks are often employed to study interval timing in humans and animals. Given the systemic delivery of pharmacological interventions, it is difficult to conclude that the effects are specific to the dorsomedial striatum. Future studies should use the local infusion of drugs into the dorsomedial striatum.

---

## [Author Response]

The following is the authors’ response to the previous reviews.

**Public Reviews:**

**Reviewer #1 (Public Review):**
Summary:In this work, the authors examine the activity and function of D1 and D2 MSNs in dorsomedial striatum (DMS) during an interval timing task. In this task, animals must first nose poke into a cued port on the left or right; if not rewarded after 6 seconds, they must switch to the other port. Thus, this task requires animals to estimate if at least 6 seconds have passed after the first nose poke. After verifying that animals estimate the passage of 6 seconds, the authors examine striatal activity during this interval. They report that D1-MSNs tend to decrease activity, while D2MSNs increase activity, throughout this interval. They suggest that this activity follows a driftdiffusion model, in which activity increases (or decreases) to a threshold after which a decision is made. The authors next report that optogenetically inhibiting D1 or D2 MSNs, or pharmacologically blocking D1 and D2 receptors, increased the average wait time. This suggests that both D1 and D2 neurons contribute to the estimate of time, with a decrease in their activity corresponding to a decrease in the rate of 'drift' in their drift-diffusion model. Lastly, the authors examine MSN activity while pharmacologically inhibiting D1 or D2 receptors. The authors observe most recorded MSNs neurons decrease their activity over the interval, with the rate decreasing with D1/D2 receptor inhibition.

We appreciate the careful read by this reviewer.

Major strengths:The study employs a wide range of techniques - including animal behavioral training, electrophysiology, optogenetic manipulation, pharmacological manipulations, and computational modeling. The question posed by the authors - how striatal activity contributes to interval timing - is of importance to the field and has been the focus of many studies and labs. This paper contributes to that line of work by investigating whether D1 and D2 neurons have similar activity patterns during the timed interval, as might be expected based on prior work based on striatal manipulations. However, the authors find that D1 and D2 neurons have distinct activity patterns. They then provide a decision-making model that is consistent with all results. The data within the paper is presented very clearly, and the authors have done a nice job presenting the data in a transparent manner (e.g., showing individual cells and animals). Overall, the manuscript is relatively easy to read and clear, with sufficient detail given in most places regarding the experimental paradigm or analyses used.

We are glad that our main points come clearly through.

Major weaknesses:One weakness to me is the impact of identifying whether D1 and D2 had similar or different activity patterns. Does observing increasing/decreasing activity in D2 versus D1, or different activity patterns in D1 and D2, support one model of interval timing over another, or does it further support a more specific idea of how DMS contributes to interval timing?

This is a great point - we were not clear. We observe distinct patterns of D2 and D1-MSN activity, but that disrupting either D2-MSNs or D1-MSNs led to increased response time. The model that this supports is that D2-MSNs and D1-MSN ensemble activity represents temporal evidence. This is a very specific model that can be rigorously tested in future work. We have now made this very clear in the abstract (Page 2).

“We found that D2-MSNs and D1-MSNs exhibited distinct dynamics over temporal intervals as quantified by principal component analyses and trial-by-trial generalized linear models. MSN recordings helped construct and constrain a fourparameter drift-diffusion computational model in which MSN ensemble activity represented the accumulation of temporal evidence. This model predicted that disrupting either D2-MSNs or D1-MSNs would increase interval timing response times and alter MSN firing. In line with this prediction, we found that optogenetic inhibition or pharmacological disruption of either D2-MSNs or D1-MSNs increased interval timing response times.”

And in the results on Page 18:

“Because both D2-MSNs and D1-MSNs accumulate temporal evidence, disrupting either MSN type in the model changed the slope. The results were obtained by simultaneously decreasing the drift rate *D* (equivalent to lengthening the neurons’ integration time constant) and lowering the level of network noise 𝝈: *D* = 𝟎. 𝟏𝟐𝟗, 𝝈 = 𝟎. 𝟎𝟒𝟑 for D2-MSNs in Fig 4A (in red; changes in noise had to accompany changes in drift rate to preserve switch response time variance. See Methods); and 𝑫 = 𝟎. 𝟏𝟐𝟐, 𝝈 = 𝟎. 𝟎𝟒𝟑 for D1-MSNs in Fig 4B (in blue). The model predicted that disrupting either D2-MSNs or D1-MSNs would increase switch response times (Fig 4C and Fig 4D) and would shift MSN dynamics.”

And in the discussion (Page 30):

“Striatal MSNs are critical for temporal control of action (Emmons et al., 2017; Gouvea et al., 2015; Mello et al., 2015). Three broad models have been proposed for how striatal MSN ensembles represent time: (1) the striatal beat frequency model, in which MSNs encode temporal information based on neuronal synchrony Matell and Meck, 2004; (2) the distributed coding model, in which time is represented by the state of the network Paton and Buonomano, 2018; and (3) the DDM, in which neuronal activity monotonically drifts toward a threshold after which responses are initiated (Emmons et al., 2017; Simen et al., 2011; Wang et al., 2018). While our data do not formally resolve these possibilities, our results show that D2-MSNs and D1MSNs exhibit opposing changes in firing rate dynamics in PC1 over the interval. Past work by our group and others has demonstrated that PC1 dynamics can scale over multiple intervals to represent time (Emmons et al., 2020, 2017; Gouvea et al., 2015; Mello et al., 2015; Wang et al., 2018). We find that low-parameter DDMs account for interval timing behavior with both intact and disrupted striatal D2- and D1-MSNs. While other models can capture interval timing behavior and account for MSN neuronal activity, our model does so parsimoniously with relatively few parameters (Matell and Meck, 2004; Paton and Buonomano, 2018; Simen et al., 2011). We and others have shown previously that ramping activity scales to multiple intervals, and DDMs can be readily adapted by changing the drift rate (Emmons et al., 2017; Gouvea et al., 2015; Mello et al., 2015; Simen et al., 2011). Interestingly, decoding performance was high early in the interval; indeed, animals may have been focused on this initial interval (Balci and Gallistel, 2006) in making temporal comparisons and deciding whether to switch response nosepokes.”

Regarding the reviewer’s specific question – it is not clear why D1-MSNs and D2-MSNs have opposing patterns of activity, as integration of temporal evidence can certainly be achieved increasing or decreasing firing rates alone. These patterns have been seen in motor control. Prefrontal neurons, which control striatal ramping, also ramp up and down. We have now included a paragraph on Page 30 explicitly discussing these ideas; however, future experiments will be required to investigate the source of the divergent patterns of activity among D2-MSNs and D1-MSNs.

“D2-MSNs and D1-MSNs play complementary roles in movement. For instance, stimulating D1-MSNs facilitates movement, whereas stimulating D2-MSNs impairs movement (Kravitz et al., 2010). Both populations have been shown to have complementary patterns of activity during movements with MSNs firing at different phases of action initiation and selection (Tecuapetla et al., 2016). Further dissection of action selection programs reveals that opposing patterns of activation among D2MSNs and D1-MSNs suppress and guide actions, respectively, in the dorsolateral striatum (Cruz et al., 2022). A particular advantage of interval timing is that it captures a cognitive behavior within a single dimension — time. When projected along the temporal dimension, it was surprising that D2-MSNs and D1-MSNs had opposing patterns of activity. Ramping activity in the prefrontal cortex can increase or decrease; and prefrontal neurons project to and control striatal ramping activity (Emmons et al., 2020, 2017; Wang et al., 2018). It is possible that differences in D2MSNs and D1-MSNs reflect differences in cortical ramping, which may themselves reflect more complex integrative or accumulatory processes. Further experiments are required to investigate these differences. Past pharmacological work from our group and others has shown that disrupting D2- or D1-MSNs slows timing (De Corte et al., 2019b; Drew et al., 2007, 2003; Stutt et al., 2024) and are in agreement with pharmacological and optogenetic results in this manuscript. Computational modeling predicted that disrupting either D2-MSNs or D1-MSNs increased selfreported estimates of time, which was supported by both optogenetic and pharmacological experiments.”

I found the results presented in Figures 2 and 3 to be a little confusing or misleading. In Figure 2, the authors appear to claim that D1 neurons decrease their activity over the time interval while D2 neurons increase activity. The authors use this result to suggest that D1/D2 activity patterns are different. In Figure 3, a different analysis is done, and this time D2 neurons do not significantly increase their activity with time, conflicting with Figure 2. While in both figures, there is a significant difference between the mean slopes across the population, the secondary effect of positive/negative slope for D2/D1 neurons changes. I find this especially confusing as the authors refer back to the positive/negative slope for D2/D1 neurons result throughout the rest of the text.

We were not clear. First, we attempted to quantify these differences based on PCA and slope. We have rephrased our characterization of these differences by changing text on (Page 9) to:

“These PETHs revealed that for the 6-second interval immediately after trial start, many putative D2-MSN neurons appeared to ramp up while many putative D1-MSNs appeared to ramp down. For 32 putative D2-MSNs average PETH activity increased over the 6-second interval immediately after trial start, whereas for 41 putative D1-MSNs, average PETH activity decreased. Accordingly, D2-MSNs and D1-MSNs had differences in activity early in the interval (0-5 seconds; F = 4.5, *p* = 0.04 accounting for variance between mice) but not late in the interval (5-6 seconds; F = 1.9, *p =* 0.17 accounting for variance between mice). Examination of a longer interval of 10 seconds before to 18 seconds after trial start revealed the greatest separation in D2-MSN and D1-MSN dynamics during the 6-second interval after trial start (Fig S2). Strikingly, these data suggest that D2-MSNs and D1-MSNs might display distinct dynamics during interval timing.”

We have rephrased our discussion on PCA to quantify differences in Fig 2G-H using data-driven methods (Page 12):

“To quantify differences between D2-MSNs vs D1-MSNs in Fig 2G-H, we turned to principal component analysis (PCA), a data-driven tool to capture the diversity of neuronal activity (Kim et al., 2017a). Work by our group and others has uniformly identified PC1 as a linear component among corticostriatal neuronal ensembles during interval timing (Bruce et al., 2021; Emmons et al., 2020, 2019, 2017; Kim et al., 2017a; Narayanan et al., 2013; Narayanan and Laubach, 2009; Parker et al., 2014; Wang et al., 2018). We analyzed PCA calculated from all D2-MSN and D1MSN PETHs over the 6-second interval immediately after trial start. PCA identified time-dependent ramping activity as PC1 (Fig 3A), a key temporal signal that explained 54% of variance among tagged MSNs (Fig 3B; variance for PC1 p = 0.009 vs 46 (44-49)% for any pattern of PC1 variance derived from random data; Narayanan, 2016). Consistent with population averages from Fig 2G&H, D2-MSNs and D1-MSNs had opposite patterns of activity with negative PC1 scores for D2MSNs and positive PC1 scores for D1-MSNs (Fig 3C; PC1 for D2-MSNs: -3.4 (-4.6 – 2.5); PC1 for D1-MSNs: 2.8 (-2.8 – 4.9); F = 8.8, p = 0.004 accounting for variance between mice (Fig S3A); Cohen’s d = 0.7; power = 0.80; no reliable effect of sex (F = 0.44, p = 0.51) or switching direction (F = 1.73, p = 0.19)).”

And finally, we directly investigate the heart of the reviewer’s question by explicitly comparing PC1 scores – a data-driven analysis of neuronal patterns that explain the least variance – and show that they are less than 0 for D2-MSNs (i.e., negatively correlated with a down-ramping pattern, or ramping up), and greater than 0 for D1MSNs (i.e., positively correlated with an up-ramping pattern):

“Importantly, PC1 scores for D2-MSNs were significantly less than 0 (signrank D2MSN PC1 scores vs 0: p = 0.02), implying that because PC1 ramps down, D2-MSNs tended to ramp up. Conversely, PC1 scores for D1-MSNs were significantly greater than 0 (signrank D1-MSN PC1 scores vs 0: p = 0.05), implying that D1-MSNs tended to ramp down. Thus, analysis of PC1 in Fig 3A-C suggested that D2-MSNs (Fig 2G) and D1-MSNs (Fig 2H) had opposing ramping dynamics.”

We interpret these data on Page 16:

“Our analysis of average activity (Fig 2G-H) and PC1 (Fig 3A-C) suggested that D2MSNs and D1-MSNs might have opposing dynamics. However, past computational models of interval timing have relied on drift-diffusion dynamics that increases over the interval and accumulates evidence over time (Nguyen et al., 2020; Simen et al., 2011).”

The reviewer mentions our analysis of ‘mean slopes across the population’ -which we clarify as trial-by-trial slope analysis, which is distinct from the population averages in 2G-H and 3A-C. We have now made this clear (Page 12).

“To interrogate these dynamics at a trial-by-trial level, we calculated the linear slope of D2-MSN and D1-MSN activity over the first 6 seconds of each trial using generalized linear modeling (GLM) of effects of time in the interval vs trial-by-trial firing rate (Latimer et al., 2015). Note that this analysis focuses on each trial rather than population averages in Fig 2G-H and Fig 3A-C.”

Finally, as the reviewer suggests, we have removed the term ‘slope’ from the rest of the paper, as the increasing/decreasing comes from averages and analyses of PC1. We have removed all discussion of ‘opposing’ slope or ‘increasing/decreasing’ slope.

It is a bit unclear to me how the authors chose the parameters for the model, and how well the model explains behavior is quantified. It seems that the authors didn't perform cross-validation across trials (i.e., they chose parameters that explained behavior across all trials combined, rather than choosing parameters from a subset of trials and determining whether those parameters are robust enough to explain behavior on held-out trials). I think this would increase the robustness of the result.In addition, it remains a bit unclear to me how the authors changed the specific parameters they did to model the optogenetic manipulation. It seems these parameters were chosen because they fit the manipulation data. This makes me wonder if this model is flexible enough that there is almost always a set of parameters that would explain any experimental result; in other words, I'm not sure this model has high explanatory power.

We are glad the reviewer raised these points. First, we have now included a complete exploration of the parameter space, exactly as the reviewer recommends. These are described in the methods (Page 41):

“Selection of DDMs parameters. Our goal was to build DDMs with dynamics that produce “response times” according to the observed distribution of mice switch times. The selection of parameter values in Fig 4 was done in three steps. First, we fit the distribution of the mice behavioral data with a Gamma distribution and found its fitting values for shape 𝜶𝑴 and rate 𝜷𝑴 (Table S2 and Fig S8; R2 Data vs Gamma ≥ 𝟎. 𝟗𝟒). We recognized that the mean 𝝁𝑴 and the coefficient of variation 𝑪𝑽𝑴 are directly related to the shape and rate of the Gamma distribution by formulas 𝝁𝑴 = 𝜶𝑴/𝜷𝑴 and 𝑪𝑽𝑴 = 𝟏/√𝜶𝑴. Next, we fixed parameters 𝑭 and 𝒃 in DDM (e.g., for D2-MSNs: 𝑭 = 𝟏, 𝒃 = 𝟎. 𝟓𝟐) and simulated the DDM for a range of values for 𝑫 and 𝝈. For each pair (𝑫, 𝝈), one computational “experiment” generated 500 response times with mean 𝝁 and coefficient of variation 𝑪𝑽. We repeated the “experiment” 10 times and took the group median of 𝝁 and 𝑪𝑽 to obtain the simulation-based statistical measures 𝝁𝑺 and 𝑪𝑽𝑺. Last, we plotted 𝑬𝝁 = |(𝝁𝑺 − 𝝁𝑴)/𝝁𝑴| and 𝑬𝒄𝒗 = |𝑪𝑽𝑺 − 𝑪𝑽𝑴|, the respective relative error and the absolute error to data (Fig S7). We considered that parameter values (𝑫, 𝝈) provided a good DDM fit of mice behavioral data whenever 𝑬𝝁 ≤ 𝟎. 𝟎𝟓 and 𝑬𝒄𝒗

And included a new Fig S7 which shows the parameter space:

These new data clearly comment on the parameter space of our model.

Finally, the reviewer mentions cross-validation. We did this at length on our model and data fits. We used 10-fold cross-validation as fitlm needs enough data for the individual fits. We found that the fit was extremely stable – i.e, we ended up with standard deviations in R2<0.004 for all comparisons. Thus, we added the following point to the methods on Page 41:

“10-fold cross-validation revealed highly stable fits between gamma, models and data.”

Lastly, the results are based on a relatively small dataset (tens of cells).

This is an important point. Although it is a small optogenetically-tagged dataset, we have adequate statistical power and large effect sizes, which we now detail in the text on Page 12:

“Consistent with population averages from Fig 2G&H, D2-MSNs and D1-MSNs had opposite patterns of activity with negative PC1 scores for D2-MSNs and positive PC1 scores for D1-MSNs (Fig 3C; PC1 for D2-MSNs: -3.4 (-4.6 – 2.5); PC1 for D1MSNs: 2.8 (-2.8 – 4.9); F = 8.8, p = 0.004 accounting for variance between mice (Fig S3A); Cohen’s d = 0.7; power = 0.80; no reliable effect of sex (F = 0.44, p = 0.51) or switching direction (F = 1.73, p = 0.19)).”

And:

“GLM analysis also demonstrated that D2-MSNs had significantly different slopes (0.01 spikes/second (-0.10 – 0.10)), which were distinct from D1-MSNs (-0.20 -0.47– 0.06; Fig 3D; F = 8.9, p = 0.004 accounting for variance between mice (Fig S3B); Cohen’s d = 0.8; power = 0.98; no reliable effect of sex (F = 0.02, p = 0.88) or switching direction (F = 1.72, p = 0.19)).”

And we have included the reviewers point as a limitation on Page 33:

“Second, although we had adequate statistical power and medium-to-large effect sizes, optogenetic tagging is low-yield, and it is possible that recording more of these neurons would afford greater opportunity to identify more robust results and alternative coding schemes, such as neuronal synchrony.”

Impact:The task and data presented by the authors are very intriguing, and there are many groups interested in how striatal activity contributes to the neural perception of time. The authors perform a wide variety of experiments and analysis to examine how DMS activity influences time perception during an interval-timing task, allowing for insight into this process. However, the significance of the key finding -- that D1 and D2 activity is distinct across time -- remains somewhat ambiguous to me.

Again, we are glad that the reviewer appreciated our main point, and we very much appreciate the additional points about interpretation, model parameters, and statistical power. If there is any way we can clarify the text further we are happy to do so.

**Reviewer #2 (Public Review):**
(1) Regarding the results in Figure 2 and Figure 5: for the heatmaps in Fig.2F and Fig.2E, the overall activity pattern of D1 and D2 MSNs looks very similar, both D1 and D2 MSNs contains neurons showing decreasing or increasing activity during interval timing. And the optogenetic and pharmacologic inhibition of either D1 or D2 MSNs resulted in similar behavior outcomes. To me, the D1 and D2 MSN activities were more complementary than opposing.

This is a great point. In our last revision, R3 suggested that complementary means opposing – and suggested we change the title to reflect this. Our original title was ‘Complementary cognitive roles for D2-MSNs and D1-MSNs during interval timing’ – and we have changed the title back to this. We have clarified what we meant by complementary in the abstract (Page 2):

“Together, our findings demonstrate that D2-MSNs and D1-MSNs had opposing dynamics yet played complementary cognitive roles, implying that striatal direct and indirect pathways work together to shape temporal control of action.”

And on Page 30:

“These data, when combined with our model predictions, demonstrate that despite opposing dynamics, D2-MSNs and D1-MSN contribute complementary temporal evidence to controlling actions in time.”

If the authors want to emphasize the opposing side of D1 and D2 MSNs, then the manipulation experiments need to be re-designed, since the average activity of D2 MSNs increased, while D1 MSNs decreased during interval timing, instead of using inhibitory manipulations in both pathways, the authors should use inhibitory manipulation in D2-MSNs, while using optogenetic or pharmacology to activate D1-MSNs. In this way, the authors can demonstrate the opposing role of D1 and D2 MSNs and the functions of increased activity in D2-MSNs and decreased activity in D1-MSNs.

These are great ideas, which we agree with. We would like to emphasize the complementary nature as noted in our original title, and not the opposing side of D1/D2 MSNs. The experiments proposed by reviewer are certainly worth doing, but would likely be quite complex to find the right stimulation parameters to affect timing without affecting movement – and we have now included them as an important limitation / future direction (Page 33):

“Fifth, we did not deliver stimulation to the striatum because our pilot experiments triggered movement artifacts or task-specific dyskinesias (Kravitz et al., 2010). Future stimulation approaches carefully titrated to striatal physiology may affect interval timing without affecting movement.”

(2) Regarding the results in Figure 3 C and D, Figure 6 H and Figure 7 D, what is the sample size? From the single data points in the figures, it seems that the authors were using the number of cells to do statistical tests and plot the figures. For example, Figure 3 C, if the authors use n = 32 D2 MSNs and n = 41D1 MSNs to do the statistical test, it could make a small difference to be statistically significant. The authors should use the number of mice to do the statistical tests.

These are important points that were discussed at length in the prior review. First, for the sample size, we now have detailed in our Table 1:

Second, we have detailed our statistical approach which explicitly deals with repeated observations of neurons across mice (Page 43):

“Statistics. All data and statistical approaches were reviewed by the Biostatistics, Epidemiology, and Research Design Core (BERD) at the Institute for Clinical and Translational Sciences (ICTS) at the University of Iowa. All code and data are made available at http://narayanan.lab.uiowa.edu/article/datasets. We used the median to measure central tendency and the interquartile range to measure spread. We used Wilcoxon nonparametric tests to compare behavior between experimental conditions and Cohen’s d to calculate effect size. Analyses of putative single-unit activity and basic physiological properties were carried out using custom routines for MATLAB. For all neuronal analyses, variability between animals was accounted for using generalized linear-mixed effects models and incorporating a random effect for each mouse into the model, which allows us to account for inherent betweenmouse variability. We used fitglme in MATLAB and verified main effects using lmer in R. We accounted for variability between MSNs in pharmacological datasets in which we could match MSNs between saline, D2 blockade, and D1 blockade. P values < 0.05 were interpreted as significant.”

We have formally reviewed this approach with professional biostatisticians at the University of Iowa.

Finally, we note that we do have adequate statistical power for analysis of Fig 3C and D: we have adequate statistical power and large effect sizes, which we now detail in the text on Page 12:

“Consistent with population averages from Fig 2G&H, D2-MSNs and D1-MSNs had opposite patterns of activity with negative PC1 scores for D2-MSNs and positive PC1 scores for D1-MSNs (Fig 3C; PC1 for D2-MSNs: -3.4 (-4.6 – 2.5); PC1 for D1MSNs: 2.8 (-2.8 – 4.9); F = 8.8, p = 0.004 accounting for variance between mice (Fig S3A); Cohen’s d = 0.7; power = 0.80; no reliable effect of sex (F = 0.44, p = 0.51) or switching direction (F = 1.73, p = 0.19)).”

And, on Page 12:

“GLM analysis also demonstrated that D2-MSNs had significantly different slopes (0.01 spikes/second (-0.10 – 0.10)), which were distinct from D1-MSNs (-0.20 -0.47– 0.06; Fig 3D; F = 8.9, p = 0.004 accounting for variance between mice (Fig S3B); Cohen’s d = 0.8; power = 0.98; no reliable effect of sex (F = 0.02, p = 0.88) or switching direction (F = 1.72, p = 0.19)).”

And we have included the reviewers point as a limitation on Page 33:

“Second, although we had adequate statistical power and medium-to-large effect sizes, optogenetic tagging is low-yield, and it is possible that recording more of these neurons would afford greater opportunity to identify more robust results and alternative coding schemes, such as neuronal synchrony.”

(3) Regarding the results in Figure 5, wly at is the reason for the increase in the response times? The authors should plot the position track during intervals (0-6 s) with or without optogenetic or pharmacologic inhibition. The authors can check Figures 3, 5, and 6 in the paper https://doi.org/10.1016/j.cell.2016.06.032 for reference to analyze the data.

These are key points, and we are glad the reviewer raised them. Our interpretation is that response time increases – without reliable changes in other task-specific movements such as nosepoke reaction time or traversal time (Fig S9). This was lacking in our prior manuscript, and we are glad the reviewer raised it. We have now added this to Page 30

“Our interpretation is that because the activity of D2-MSN and D1-MSN ensembles represents the accumulation evidence, pharmacological/optogenetic disruption of D2-MSN/D1-MSN activity slows this accumulation process, leading to slower interval timing-response times (Fig 5) without changing other task-specific movements (Fig S9). These results provide new insight into how opposing patterns of striatal MSN activity control behavior in similar ways and show that they play a complementary role in elementary cognitive operations.”

Regarding the tracking of velocity, we unfortunately do not have this information reliably across all conditions. This citation is a beautiful landmark paper, and we are working on collecting this information in our new datasets going forward. We have included this as a major limitation (Page 34):

“Still, future work combining motion tracking/accelerometry with neuronal ensemble recording and optogenetics and including bisection tasks may further unravel timing vs. movement in MSN dynamics (Robbe, 2023; Tecuapetla et al., 2016).”

Once again, we are appreciative of the thoughtful points raised by this reviewer.

**Reviewer #3 (Public Review):**
Summary:The cognitive striatum, also known as the dorsomedial striatum, receives input from brain regions involved in high-level cognition and plays a crucial role in processing cognitive information. However, despite its importance, the extent to which different projection pathways of the striatum contribute to this information processing remains unclear. In this paper, Bruce et al. conducted a study using various causal and correlational techniques to investigate how these pathways collectively contribute to interval timing in mice. Their results were consistent with previous research, showing that the direct and indirect striatal pathways perform opposing roles in processing elapsed time. Based on their findings, the authors proposed a revised computational model in which two separate accumulators track evidence for elapsed time in opposing directions. These results have significant implications for understanding the neural mechanisms underlying cognitive impairment in neurological and psychiatric disorders, as disruptions in the balance between direct and indirect pathway activity are commonly observed in such conditions.Strengths:The authors employed a well-established approach to study interval timing and employed optogenetic tagging to observe the behavior of specific cell types in the striatum. Additionally, the authors utilized two complementary techniques to assess the impact of manipulating the activity of these pathways on behavior. Finally, the authors utilized their experimental findings to enhance the theoretical comprehension of interval timing using a computational model.

We very much appreciate the considered read and comments by the reviewer, and recognition of the breadth of techniques in this manuscript.

Weaknesses:The behavioral task used in this study is best suited for investigating elapsed time perception, rather than interval timing. Timing bisection tasks are often employed to study interval timing in humans and animals. In the optogenetic experiment, the laser was kept on for too long (18 seconds) at high power (12 mW). This has been shown to cause adverse effects on population activity (for example, through heating the tissue) that are not necessarily related to their function during the task epochs. Given the systemic delivery of pharmacological interventions, it is difficult to conclude that the effects are specific to the dorsomedial striatum. Future studies should use the local infusion of drugs into the dorsomedial striatum.

These are important points. We agree with them completely and have now included responses to them. First, bisection tasks certainly have advantages – we have justified our approach in the discussion (Page 32):

“Our task version has been used extensively to study interval timing in mice and humans (Balci et al., 2008; Bruce et al., 2021; Stutt et al., 2024; Tosun et al., 2016; Weber et al., 2023). However, temporal bisection tasks, in which animals hold during a temporal cue and respond at different locations depending on cue length, have advantages in studying how animals time an interval because animals are not moving while estimating cue duration (Paton and Buonomano, 2018; Robbe, 2023; Soares et al., 2016). Our interval timing task version – in which mice switch between two response nosepokes to indicate their interval estimate has elapsed – has been used extensively in rodent models of neurodegenerative disease (Larson et al., 2022; Weber et al., 2024, 2023; Zhang et al., 2021), as well as in humans (Stutt et al., 2024). This version of interval timing involves motor timing, which engages executive function and has more translational relevance for human diseases than perceptual timing or bisection tasks (Brown, 2006; Farajzadeh and Sanayei, 2024; Nombela et al., 2016; Singh et al., 2021). Furthermore, because many therapeutics targeting dopamine receptors are used clinically, these findings help describe how dopaminergic drugs might affect cognitive function and dysfunction. Future studies of D2-MSNs and D1-MSNs in temporal bisection and other timing tasks may further clarify the relative roles of D2- and D1-MSNs in interval timing and time estimation.”

Second – we have included an explicit control that has the same laser that is on for the same epoch as in the experimental animal – and find no effects. This is now detailed in the methods: (Page 37):

“To control for heating and nonspecific effects of optogenetics, we performed control experiments in mice without opsins using identical laser parameters in D2-cre or D1-cre mice (Fig S6).”

And in the results (Page 21):

“To control for heating and nonspecific effects of optogenetics, we performed control experiments in D2-cre mice without opsins using identical laser parameters; we found no reliable effects for opsin-negative controls (Fig S6).”

And on Page 21:

“As with D2-MSNs, we found no reliable effects with opsin-negative controls in D1MSNs (Fig S6).”

We have now detailed these results in Figure S6:

Regarding focal pharmacology, we performed this experiment with focal infusion of D1/D2 antagonists in our prior work, which we have now cited (Page 4):

“Similar behavioral effects were found with systemic (Stutt et al., 2024) or focal infusion of D2 or D1 antagonists locally within the dorsomedial striatum (De Corte et al., 2019a).”

Comments on revised version:Thank you for the comprehensive revisions. Most of my (addressable) concerns were addressed. The current version of your manuscript appears significantly improved.

Once again, we appreciate the reviewer’s constructive and insightful comments and careful review of our manuscript. Their comments have been extremely helpful.